# Combining modelled snowpack stability with machine learning to predict avalanche activity

Léo Viallon-Galinier[1,2,3], Pascal Hagenmuller[1], and Nicolas Eckert[2]

[1]Univ. Grenoble Alpes, Université de Toulouse, Météo-France, CNRS, CNRM, Centre d'Études de la Neige, Grenoble, France
[2]Univ. Grenoble Alpes, INRAE, CNRS, IRD, Grenoble INP, IGE, 38000 Grenoble, France
[3]École des Ponts, Champs-sur-Marne, France

**Correspondence:** Léo Viallon-Galinier (leo.viallon@meteo.fr)

**Abstract.** Predicting avalanche activity from meteorological and snow cover simulations is critical in mountainous areas to support operational forecasting. Several numerical and statistical methods have tried to address this issue. However, it remains unclear how combining snow physics, mechanical analysis of snow profiles and observed avalanche data improves avalanche activity prediction. This study combines extensive snow cover and snow stability simulations with observed avalanche occurrences within a Random Forest approach to predict avalanche situations at a spatial resolution corresponding to elevations and aspects of avalanche paths in a given mountain range. We develop a rigorous leave-one-out evaluation procedure including an independent evaluation set, confusion matrices, and receiver operating characteristic curves. In a region of the French Alps (Haute-Maurienne) and over the period 1960-2018, we show the added value within the machine learning model of considering advanced snow cover modelling and mechanical stability indices instead of using only simple meteorological and bulk information. Specifically, using mechanically-based stability indices and their time derivatives in addition to simple snow and meteorological variables increases the probability of avalanche situation detection from around 65% to 76%. However, due to the scarcity of avalanche events and the possible misclassification of non-avalanche situations in the training data set, the predicted avalanche situations that are really observed remains low, around 3.3%. These scores illustrate the difficulty of predicting avalanche occurrence with a high spatio-temporal resolution, even with the current data and modelling tools. Yet, our study opens perspectives to improve modelling tools supporting operational avalanche forecasting.

**Keywords:** snow avalanche, machine learning, mechanical stability indices, snow cover modelling, cross-validation, avalanche forecasting

## 1 Introduction

Avalanches are a significant issue in mountain areas where they threaten recreationists and infrastructures (Wilhelm et al., 2001; Stethem et al., 2003). The mapping (Keylock et al., 1999; Eckert et al., 2010b) and forecasting (Schweizer et al., 2020) of avalanche hazard and related risks are therefore important challenges for local authorities (Bründl and Margreth, 2021; Eckert and Giacona, 2022). Most of the countries facing such hazards rely on operational services for avalanche hazard forecasting (LaChapelle, 1977; Morin et al., 2020) and hazard mapping (Eckert et al., 2018). In this work, we focus on the issue

of forecasting (estimation of the outcomes of unseen data) of daily avalanche activity from simulated meteorological and snow data. Indeed, inferring the relation between avalanche activity and given weather and snow conditions is one of the essential components of operational avalanche hazard forecasting (prediction in the future based on predicted snow and weather conditions).

Prediction of avalanche activity is mainly based on the knowledge of the snowpack evolution and of the mechanical processes leading to avalanches (e.g. LaChapelle, 1977; Morin et al., 2020). Information on the snowpack evolution can be collected through field observations and measurements (e.g. Coléou and Morin, 2018), and numerical simulations (e.g. Bartelt and Lehning, 2002; Vionnet et al., 2012). These data typically include a detailed description of the snowpack stratigraphy with vertical profiles of snow properties (Fierz et al., 2009). Several methods allow for identifying avalanche-prone situations from these profiles. Detection of weak layers based on mechanical and expert rules, such as the so-called lemons technique (Schweizer and Jamieson, 2007), comprises one qualitative approach. Numerical computation of stability indices based on mechanical theories constitutes an automated method to quantify the snowpack stability (Roch, 1966; Föhn, 1987; Lehning et al., 2004; Schweizer et al., 2006; Viallon-Galinier et al., 2021). These approaches rely on the knowledge of mechanical processes involved in avalanche release (Schweizer, 2017; Viallon-Galinier et al., 2021). Numerical models, which are currently used as an aid to decision-making for avalanche forecasters, generally combine mechanical stability indices and expert rules to provide information on snowpack stability (Morin et al., 2020; Schweizer et al., 2006; Giraud et al., 2002; Viallon-Galinier et al., 2021).

Machine learning techniques can approach the complex link between simple snow cover variables and avalanche occurrence. These methods allow taking advantage of the knowledge of past avalanche activity to determine objective delimitation of avalanche-prone conditions within the space defined by their potential drivers. The first attempt to use machine learning techniques in the avalanche community was performed using linear methods by Bois et al. (1974). In the next decade, several attempts were made to use nearest neighbors for local avalanche danger forecasting (e.g. Navarre et al., 1987; Buser, 1989). Classification trees quickly became another common choice, as it is conceptually close to decision processes used by forecasters (e.g. Kronholm et al., 2006; Hendrikx et al., 2014). The first use of random forests was performed by Mitterer and Schweizer (2013). This method became popular in the community (e.g. Sielenou et al., 2021; Pérez-Guillén et al., 2022; Sielenou et al., 2021). Other techniques have also been tested, such as support vector machine (e.g. Pozdnoukhov et al., 2011; Choubin et al., 2019; Sielenou et al., 2021) and more advanced techniques appeared in the last years such as convolutional neural networks (e.g. Singh and Ganju, 2008; Dekanova et al., 2018).

Most existing studies used meteorological variables as input or simple bulk variables such as snow depth to feed the machine learning model. The first machine learning models (Navarre et al., 1987; Buser, 1989) mainly relied on meteorological observations, simple snow observations and avalanche records. The use of modelled snow information was therefore developed to complement or replace observations (e.g. Schirmer et al., 2009; Sielenou et al., 2021) and expert analyses were introduced to provide appropriate variables (Schweizer and Föhn, 1996). However, most of the commonly used variables are only surrogates for the true drivers of avalanche processes. By contrast, studies using mechanically-based variables closely related to the processes involved in avalanche formation (e.g. Viallon-Galinier et al., 2021) are less frequent in machine learning approaches

(e.g. Schweizer and Föhn, 1996; Mayer et al., 2022). However, these variables could increase the interpretability of the algorithm results and bring complementary non-linear information readily oriented toward the prediction of avalanche activity. Hence, they may reduce the complexity of statistical tools to implement (simpler statistical relations and a smaller number of variables to consider) compared to a model that directly uses the snow model output, and improve the overall predictive power.

Existing statistical prediction approaches are difficult to compare. Different spatial extensions are considered from large mountain ranges (e.g. Kronholm et al., 2006; Sielenou et al., 2021) to avalanche paths (e.g. Choubin et al., 2019). In the literature, different measures of avalanche activity are also considered from binary classes (e.g. Kronholm et al., 2006; Hendrikx et al., 2014) to ordinal multi-classes (e.g. Mosavi et al., 2020; Sielenou et al., 2021). Yet, the most important difficulty for the comparison is that existing studies do not share a common evaluation process which includes a relevant segmentation of the training and evaluation datasets and common performance metrics. This absence of a homogeneous methodology for evaluating machine learning approaches within the snow and avalanche community limits the comparison between studies.

On this basis, this paper aims to determine whether combining machine learning on avalanche data and mechanical stability analysis of snow profiles helps predict avalanche activity. In particular, we compare the prediction score of the model trained either only on meteorological and simple snow variables as input or also on variables related to the snowpack stability and derived from the full snowpack stratigraphy. We use random forest techniques to relate meteorological, modelled snowpack information and mechanically-based stability indices to observed avalanche occurrences. We also employ time derivatives of mechanical indices to account for short-time persistence of avalanche-prone conditions in certain cases. We eventually present a rigorous leave-one-out evaluation procedure of broad interest for evaluating avalanche prediction efficiency that includes an independent evaluation set, confusion matrices, receiver operating characteristic (ROC) curves and additional scores derived from the confusion matrix. The study area is located in Haute-Maurienne in the French Alps where extensive avalanche data and snow cover reanalyses over 58 years (1960-2018) are available.

## 2 Material and methods

### 2.1 Study area

We selected an area belonging to the Haute-Maurienne massif in the Northern French Alps, consisting of the three district municipalities of Bessans, Bonneval-sur-Arc and Lanslevillard (Figure 1). This area is frequently studied for avalanche-related issues (e.g. Ancey et al., 2004; Eckert et al., 2009; Favier et al., 2014; Kern et al., 2021; Zgheib et al., 2020) because it is prone to intense avalanche activity. The area is characterized by a relatively high elevation ranging from 1400 to 3700 m, and its avalanche activity does not yet seem to be reduced by adverse climate warming effects (Lavigne et al., 2015; Zgheib et al., 2022). Located in the eastern French Alps next to the Italian border, the area experiences extreme snowfall events known as "easterly return", which drive most of the avalanche activity (Eckert et al., 2010a; Le Roux et al., 2021). We considered data on the winters between 1960 and 2018. When referring to the winter season, we consider days between the 15th of October and the 15th of May. These dates are consistent with the dates of production of avalanche bulletins in France and were already selected as suitable bounds in other studies (e.g. Sielenou et al., 2021).

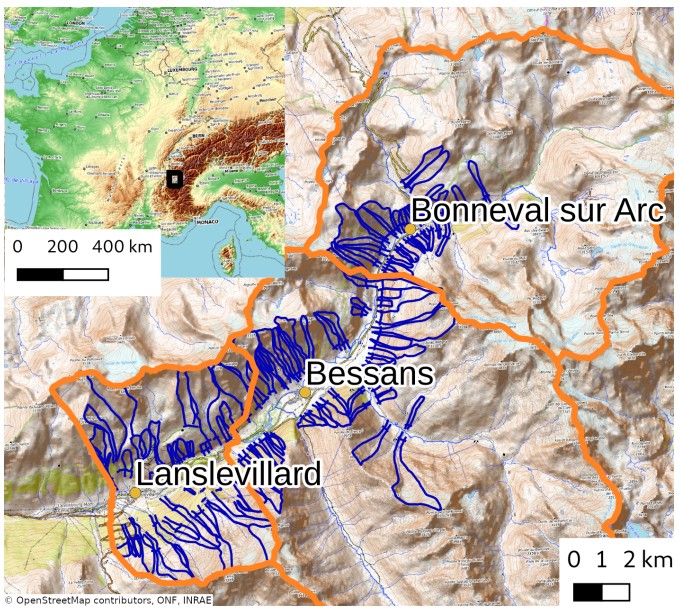

**Figure 1.** Studied area. General situation on the left and contour of avalanche paths surveyed each day (EPA) in blue for the three district municipalities of our studied area (delimited with orange lines). Only the avalanches that flow below a certain threshold (blue line at the bottom of each avalanche path) are systematically reported.

## 2.2 Avalanche observations

Our proxy of avalanche activity relies on the *Enquête Permanente sur les Avalanches* (EPA). The EPA reports all avalanches in approximately 3,000 pre-defined paths over French mountain ranges (Bourova et al., 2016). About 110 of these are located in the studied area and are shown in Figure 1. Each avalanche record indicates the period during which the avalanche is likely to have released and some additional information, such as the elevation and the aspect of the starting zone. EPA was initially designed to capture large natural avalanche events in exposed areas and was extensively used for hazard mapping (Bourova et al., 2016). Hence, only avalanches whose run out reaches a certain pre-identified run out threshold (defined for each avalanche path, with a threshold elevation, e.g.: a road, or the valley floor, see Figure 1) are systematically recorded. The avalanche activity derived from the EPA depends on this specific sampling procedure. Moreover, it relies on human-based observations and inevitably contains some uncertainties. However, EPA remains one of the longest avalanche activity records. The selected area is characterized by a dense observation network covering a large variety of avalanche paths. Besides, the steep topography of Haute-Maurienne reduces the effect of the observation threshold as most avalanches flow far downslope, close to the valley floor. Further discussion on the EPA strengths and weaknesses is out of the scope of the paper and can be found in Jomelli et al. (2007) or Eckert et al. (2013).

One of the drawbacks of this data for the current study is the uncertainty of the date of some avalanche events, which can be large for remote paths or during low visibility periods (28.6% of the reports have an uncertainty above one day and 23.6%

above 3 days, as estimated by the observers). To associate meteorological and snow conditions to each observed avalanche, we remove observations with an uncertainty (length of the period on which the avalanche can have occurred) of more than three days on the release date, from the dataset. When the uncertainty is larger than one day, the last day of the period was defined as the day of the avalanche event. For instance, if an observer reports that an avalanche has occurred between the 21st and 23rd of January in a given path, we consider that the uncertainty of the report is 3 days (<= 3 days) and we arbitrarily consider that the avalanche occurred on the 23rd of January. Moreover, the aspect and the elevation of the starting zone were not reported in a few cases (representing less than 5% of the total number of events) because the starting point was not visible from the observation point or due to a lack of time for the observation. In these cases, the starting zone was defined by the average elevation and aspect of the typical release area defined for each avalanche path. We applied this definition of release day and zone to the 2779 observed avalanches in the studied domain.

We grouped these observations into eight aspect sectors (from North to North-West) and three elevation bands (centered at 1800, 2400 and 3000 m). This choice defines the spatial resolution of our model. All observations are represented in this geometry in Figure 2. When considering all avalanche and non-avalanche situations, the avalanche situations represent 1.1% of the overall dataset. This is called the base rate and acts as a reference for further comparisons.

### 2.3  Simulated snowpack

The SAFRAN-SURFEX/ISBA-Crocus model chain (Durand et al., 1999; Lafaysse et al., 2013) was used to simulate the snow and meteorological conditions in the Haute-Maurienne massif. SAFRAN provides meteorological information adapting numerical weather prediction on a gridded domain to the area of interest and assimilates observed meteorological data (Durand et al., 2009). We used the publicly available reanalysis (Vernay et al., 2020). This modelling scheme assumes that meteorological conditions depend only on elevation and aspect. The SURFEX/ISBA-Crocus model is a one-dimensional snowpack model representing snowpack evolution with a multi-layered scheme based on physical evolution laws (Brun et al., 1989; Vionnet et al., 2012). It uses as an input the meteorological data from SAFRAN model, and it is coupled to the soil scheme ISBA-DIF (Decharme et al., 2011) to represent energy and mass exchange at the bottom of the snowpack. Accordingly to the spatial resolution of the avalanche observations, snow conditions are computed for eight aspects and three elevation levels (1800, 2400 and 3000 m). The temporal resolutions of the meteorological and snow conditions considered here were 1 h and 3 h, respectively.

These simulations retrieve meteorological and bulk snow conditions but also the full snowpack stratigraphy. Hence an additional step is required to take advantage of this information, which is here done through the computation of stability indices as presented right after.

### 2.4  Stability indices

Nine stability indices have been selected based on their applicability with our snow cover model: five for dry snow avalanches and four for wet snow avalanches. In addition, we computed time derivatives of these indices. We also introduce the snow depth as an indicator of the amount of snow that can be involved in a potential avalanche.

### 2.4.1 Dry snow indices

For dry snow, three indices are related to failure initiation, namely natural strength-stress ratio ($S_n$, (Föhn, 1987)), skier strength-stress ratio ($S_a$, (Föhn, 1987)) and external strength-stress ratio ($S_r$, (Reuter et al., 2015)). These indices compare shear strength to shear stress, for a given layer interface, where the stress originates from the weight of the overlying layers ($S_n$ and $S_a$) and/or of an external load (skier, for instance) at the top of the snowpack ($S_a$ and $S_r$) (Viallon-Galinier et al., 2021). Moreover, we selected two formulations of critical crack length for representing crack propagation (Viallon-Galinier et al., 2021): the original formulation by (Heierli et al., 2008; van Herwijnen et al., 2016) and the alternative approach by Gaume et al. (2017). Both approaches require a slab modulus, determined from density according to Scapozza (2004), and fracture energy estimated from strength. Details on these indices are available in Viallon-Galinier et al. (2022).

These indices were computed for each layer. For each time step, based on the values of each index, we identified five weak layers (one per index). We defined a weak layer as a layer characterized by a local minimum of the considered stability index (excluding the top and the bottom layers). This approach allows identifying the five weakest layers, with five complementary ways of estimating the weakness (five stability indices). It has the advantage of providing a constant number of variables (25 variables: 5 stability indices on 5 weak layers) for further statistical analysis.

### 2.4.2 Wet snow indices

To characterize the conditions prone to wet snow avalanches, we used the mean liquid water content in the whole snowpack (Mitterer et al., 2013, 2016) and the thicknesses of humid snow layers. For the latter index, we considered a snow layer as humid as soon as its liquid water content exceeds either 0, 1 and 3% in volume. These three indices are denoted $I_{h0}$, $I_{h1}$ and $I_{h3}$.

### 2.4.3 Time derivatives

Stability indices at a given time may not be sufficient to represent the avalanche activity. The time evolution of snow properties is supposed to be represented by snow cover models. However, considering snowpack properties only at a given date and disregarding its past evolution does not indicate whether the snowpack is becoming more prone to avalanches or is in a stabilization phase. For instance, low values of a stability index may indicate an avalanche-prone situation. However, if these values are preceded by even lower ones, the possible avalanche should already have occurred when the stability was minimal, or even before, but not after. Yet, few stability indices include the time dimension in the literature. To our knowledge, only Conway and Wilbour (1999) (also used by Reuter et al. (2022)) and MEPRA natural risk (Giraud et al., 2002; Viallon-Galinier et al., 2021) include explicit time dependence. Here, we used time derivatives of the previously defined stability indices. We defined the time derivative of stability index $f$ on a given weak layer as $(f(t) - f(t - \mathrm{d}t))/\mathrm{d}t$ with several time intervals $\mathrm{d}t$ (6, 24, 48, 72, 120 and 240 h). The derivatives represent 150 variables for dry snow indices and 24 variables for wet snow indices. Time derivatives on snow depth were used as a straightforward indicator of stability for dry snow conditions (accumulation of new snow) and wet snow conditions (settling and melting).

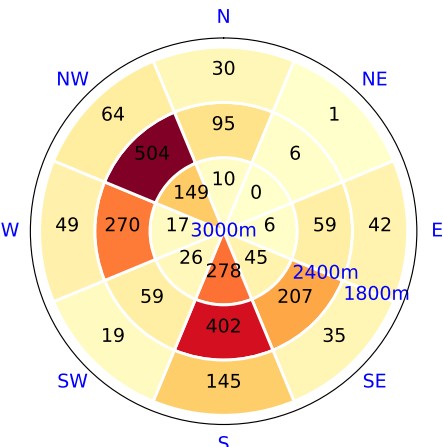

**Figure 2.** Number of avalanche situations recorded in our study area over the full time period at the presented spatial resolution, i.e. per elevation band (1800, 2400 and 3000 m) and aspects (8 from N to NW).

## 2.5 Learning procedure

Random forests were used to relate snow and meteorological conditions to avalanche activity in the presented spatial resolution.

### 2.5.1 Avalanche activity

Avalanche activity was based on EPA records in the selected area. For each day, aspect and elevation band, we classified avalanche and non-avalanche situations. A given day on given aspect sector and elevation band was considered as an avalanche situation if at least one avalanche is reported for this day (after filtering of observations and attribution of dates, as explained in Section 2.2). All other situations were non-avalanche situations. The number of avalanche situations observed by elevation and aspect range is shown in Figure 2.

### 2.5.2 General overview of input variables

For each elevation and aspect selected, input variables used are summarized in Table 1.

These variables gather information from the meteorological model SAFRAN (Meteo), SURFEX-ISBA/Crocus (Simple snow), stability indices (Stability) computed on the basis of modelled snowpack and derivatives of these variables (Derivatives) as described in Section 2.4. Hereafter, if no special mention is added, all these variables (All) were used but for studying variable 185 importance, subsets of this list are also used.

**Table 1.** Variables used to predict avalanche activity using machine learning

| Category | Sub-category | Name | time intervals | Number of variables |
|---|---|---|---|---|
| Meteo | Snowfall | Snowfall accumulation (mm) | 24 and 72h | 2 |
| | Rainfall | Rainfall accumulation (mm) | 24 and 72h | 2 |
| | Temperature | Min, max, mean values (K) | 24 and 72h | 6 |
| | Wind | Max and mean wind speed (km/h) | 24 and 72h | 4 |
| | | Projected mean direction on N-S axis and E-W axis | 24 and 72h | 4 |
| Simple snow | Snow depth | Snow depth (m) | — | 1 |
| | Depth of new snow | Depth (m) of snow fallen since (see intervals) | 24, 72, 120h | 3 |
| Stability | Dry snow | Stability indices ($S_n$, $S_a$, $S_k$, $a_c$, $a_g$) for the 5 identified weak layers and depths of each weak layer | — | 25 |
| | | Depth of the corresponding weak layers (m) | — | 5 |
| | Wet snow | Maximum mean liquid water content | 24h | 1 |
| | | Maximum height of wet snow with thresholds of 0, 1, 3% of liquid water to consider layer as wet (m) | 24h | 3 |
| | Snow depth | Snow depth | — | 1 |
| Derivatives | Dry snow indices | All dry snow indices | 6, 24, 48, 72, 120, 240h | 150 |
| | Wet snow indices | All wet snow indices | 6, 24, 48, 72, 120, 240h | 24 |
| | Snow depth variation | Snow depth variation (m) | 24, 72, 120h | 3 |

### 2.5.3   Machine learning algorithm

To relate snow and meteorological conditions to avalanche activity as defined above, we used Random Forest (RF) techniques (Breiman, 2001; Hastie et al., 2009). Random Forest is an ensemble method used for classification. Each decision tree in the ensemble is built from a random subset of the data. This technique allows going beyond the limitations of single decision trees but without dramatically increasing the algorithm complexity and with similar introspection capabilities. Once trained, each tree of the Random Forest predicts a class for the input data. Aggregating all trees allow to define a probability for each class as the portion of trees predicting the given class.

Random Forest classifiers require two hyper-parameters: the number of trees and the tree depth. Here, we let the trees fully grow until there is only one element in each leaf, as usually done (Hastie et al., 2009). An optimization on our full dataset showed that 3000 trees were sufficient (more trees did not improve the results), so that this value was selected for the whole study.

We use two classes, namely avalanche and non-avalanche situations, that are highly unbalanced (mean of 1.1% of avalanche situations in the winter season depending on elevation and aspect, see Figure 2). Machine learning techniques, if not handled with care, do not perform well on unbalanced data (e.g. Hastie et al., 2009; Sielenou et al., 2021). They are designed to optimize the overall classification accuracy or a similar score. Their results thus tend to be biased towards the majority class (Chawla et al., 2004; Sielenou et al., 2021), here the non-avalanche situations. The most common techniques to limit this effect are oversampling of the minority classes, undersampling of the majority classes or dedicated learning algorithms. We here used a combination of these techniques. We only considered situations of the winter season characterized by a simulated snow depth larger than 10 cm. This first selection led to the undersampling of the majority class. Note that we chose this conservative threshold to remove very obvious non-avalanche situations from the dataset (no snow in the starting zone means no avalanche). We do not expect this threshold to be optimal as this is the goal of the training phase of the machine learning algorithm. However, this first step was not sufficient to fully balance the dataset. We therefore used an adaptation of RF classifier to deal with unbalanced data (Chen et al., 2004): each tree of the forest is trained on a subset of the data randomly drawn; the probability law for drawing is adapted so that the probability of drawing non-avalanche or avalanche situations are identical. This second step acts as an oversampling of the minority class.

## 2.6   Evaluation methods

### 2.6.1   Evaluation process

We evaluated the model performance with a leave one year out approach (LOYO). The snowpack completely melts in summer, and new snowfall in autumn occurs on bare ground. Therefore, there is no memory between winter seasons and they are exchangeable. This is not the case between successive days during the winter season, with highly correlated snowpack characteristics. A simple leave one out (i.e., leave one day out) would yield better scores but would be less relevant. For each of the 58 seasons between 1960 and 2018, an evaluation set is composed of one winter season and a learning set of the remaining 57 seasons. This leads to 58 sets of trained random forests, each one being evaluated on one year. On a single winter season,

**Table 2.** Confusion matrix: observed and predicted avalanche situations ("Avalanche") and non-avalanche ones ("Non-avalanche").

| | | Predicted | |
|---|---|---|---|
| | | Avalanche | Non-avalanche |
| Observed | Avalanche | $AA$ | $AN$ |
| | Non-avalanche | $NA$ | $NN$ |

there are not enough avalanche situations to be statistically relevant. Therefore, the confusion matrix of 58 evaluation years were aggregated to compute scores with all information available. This leave one year out approach is used for all evaluations presented.

We also quantified the statistical uncertainty related to the sample size. As we used 58 years of evaluation data computed separately, we were able to define an uncertainty by bootstrapping evaluation years used to compute the considered score. In practice, 1000 independent draws of 58 years (with replacement) were randomly produced and the scores were computed on each draw. The 20th and 80th percentiles were used to quantify the uncertainty of the produced scores.

### 2.6.2 Scores

The Random Forest model produces the probability of being an avalanche situation, defined as a situation with at least one avalanche event, given the snow and meteorological conditions. We selected a threshold ($t$) on this probability to discriminate avalanche and non-avalanche situations. It is possible to construct a confusion matrix (as presented in Table 2) based on this threshold. We derived three scores from the confusion matrix. The true positive rate (TPR) or recall is the ratio between correctly predicted avalanche situations divided by the number of observed avalanche situations. This score is also called probability of detection (POD). It quantifies how many avalanche situations have been correctly predicted. The false positive rate (FPR), also called false alarm ratio (FAR), is the ratio between the number of false positives (non-avalanche situations that are identified as avalanche situations) and the total number of non-avalanche situations. It corresponds to the probability that a false alarm will be raised. These two complementary indicators are interesting but do not fully characterize the performance of a binary classifier in case the two classes are unbalanced (which is the case here). We used a third score to represent how many predicted avalanche situations are really observed as such. This score is called precision and is defined as the ratio between correctly predicted avalanche situations and the number of predicted avalanche situations. We also mention the specificity ($1 -$ FPR), to be compared with the true positive rate. Finally, we also compute the balanced precision, which is the precision we would have considering balanced positive and negative classes (avalanche and non avalanche situations). The definition of these scores is summarized in Table 3.

### 2.6.3 Scores presentation

These scores can be computed for any threshold $t$ on the avalanche situation probability. The impact of this threshold on the overall scores can be represented with two graphs: the ROC (Receiver Operating Characteristic) curve and the precision-recall

**Table 3.** Scores derived from the confusion matrix.

| Name | expression |
|---|---|
| True positive rate (TPR) or recall | $\frac{AA}{AA+AN}$ |
| False positive rate (FPR) | $\frac{NA}{NA+NN}$ |
| Precision | $\frac{AA}{AA+NA}$ |
| Specificity | 1 - FPR |
| Balanced precision | $\frac{AA}{AA+NA*(AA+AN)/(NA+NN)}$ |

graph. The ROC curve shows the true positive rate as a function of the false positive rate for all possible thresholds between 0 and 1. When the threshold is equal to 0, all situations are considered avalanche situations (true positive rate is 1, false positive rate is close to 1). When the threshold is 1, all situations are considered non-avalanche situations (true positive rate is 0 and false positive rate is close to zero). A perfect classifier would have a threshold value for which the true positive rate is 1, and the false positive rate is 0. Random classification is usually associated with the diagonal in the ROC diagram. A standard measure

derived from this curve is the area between the first bisector and the ROC curve (the area under curve or AUC) (Bradley, 1997). The AUC quantifies how good the model is compared to a random classifier. We used the AUC value to compare different classifier configurations. In addition, recall is also plotted as a function of precision to capture the model capacity to identify avalanche situations (precision) while limiting the number of false positives (recall). In this graph, the optimal point would be (1,1) i.e., a 100% precision and a 100% recall.

### 2.6.4   Importance of variables

The importance of variables was estimated through the separative power of each variable in the trees by computing the normalized mean decrease of impurity (also called Gini importance) on nodes where the given variable is used to separate the data in two groups (Breiman, 2001). A variable importance of zero means that the variable could be removed without reducing model performance and a high value denotes a high separative capacity (between avalanche and non-avalanche situations) of

the variable. If two variables contain similar information, each variable will be picked randomly in the tree construction and these variables will consequently share out the importance of the common information (Breiman, 2001). This first approach is commonly used with random forest but only provides a first rough insight into variable importance. We thus use a more robust discrimination of the importance of variables by using different subsets of variables (see subsets in Table 1). The performance difference between more independent groups gives an idea of the importance of the variables present in each group.

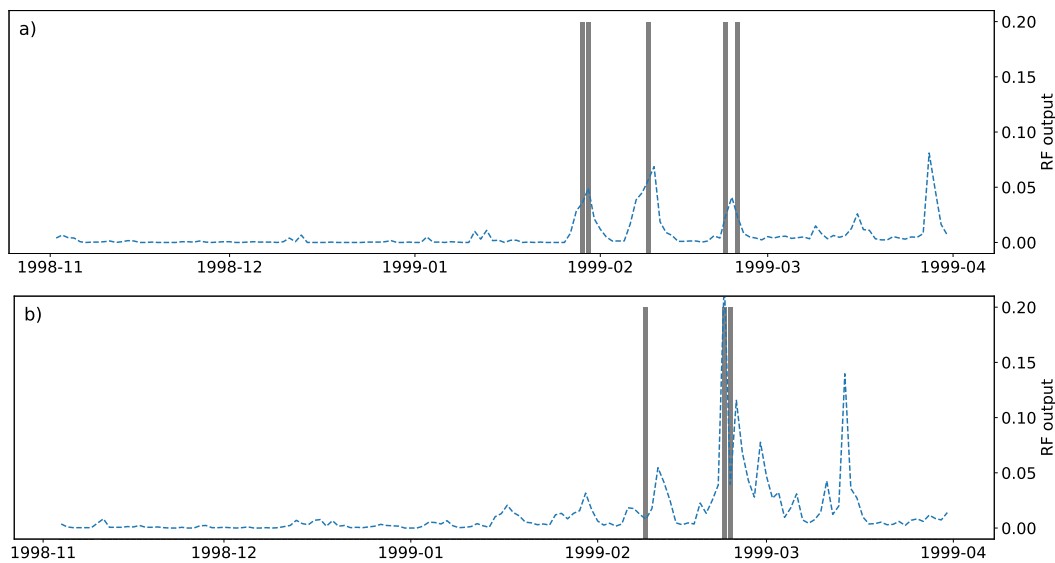

**Figure 3.** Random forest model output (trained with all variables) for winter 1998-1999 at 2400 m for aspects (a) NW and (b) SE. The grey bars represent the days for which avalanches were observed in the selected aspect and elevation range. The base rate of avalanche situations in the full dataset is 0.011. Dates represent the beginning of months.

## 3 Results

### 3.1 Overview of random forest output

The trained Random Forest model provides the probability of being an avalanche situation for each day, aspect and elevation. Figure 3 provides an overview of the output for a specific season (1998–1999), elevation (2400 m) and two aspects (NW and SE). We observe a high variability of the output between days. The time series differs between aspects, which gives a rough idea of the interest of the selected spatial scale. When considering the observations, most peaks of the random forest output correspond to observed avalanche activity. The random forest thus provides, in this example, a relevant image of the expected avalanche activity. There are also false positives (such as late March in NW aspect) or false negatives (such as early February in SE). This first overview is insufficient for an evaluation of the performance of the model that must be conducted over longer periods, all aspects and elevations.

### 3.2 Model performance

The ROC curve of the model trained with all input variables at our spatial resolution and evaluated independently on each winter season since 1958 is shown in Figure 4a. Fortunately, the model is far better than a random classifier (ROC curve above the first diagonal) but it also remains far from an optimal classifier (no points close to (0, 1)). The uncertainty around the ROC curve is very low, which indicates that a sufficient amount of data is available to constrain the model and that the evaluation is

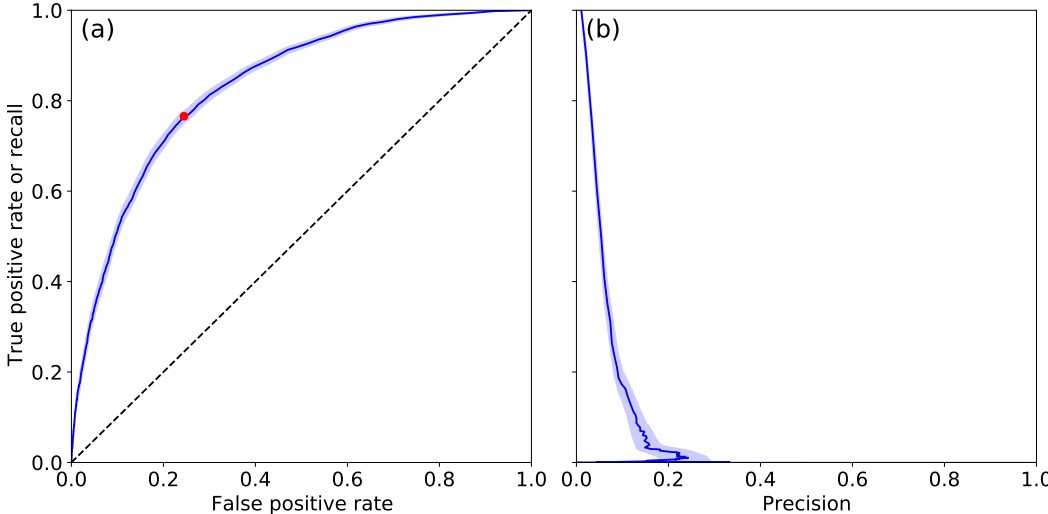

**Figure 4.** (a) ROC curve of the model trained with all input variables at our spatial resolution and evaluated independently on each winter season since 1958. The optimal point (threshold value of 0.01) is represented by a red dot. (b) Precision and recall (Table 3) curve. Shading represents the uncertainty based on the 20th and 80th percentile of the bootstrap on evaluation years (see methods section).

not sensitive to the choice of the winter season. The optimal threshold, defined as the threshold which leads to the ROC point closest to (0, 1), is here 0.01. In other words, a situation is considered an avalanche situation when the model probability is larger than 0.01. For this threshold, we provide the corresponding confusion matrix in Table 4, classifying situations between observed and predicted avalanche and non-avalanche situations in all elevation and aspect bands. The corresponding scores are 75.3% for the true positive rate or recall, 23.6% for the false positive rate and 3.3% for precision. The balanced precision is

76.2%. These scores mean that about three-quarter of the observed avalanche situations were correctly identified but avalanches were actually observed only on 3.3% of the situations when avalanches were predicted. The recall (75.3%) and sensitivity (complementary of the false positive rate, here 76.4%) are similar, indicating similar performances on observed avalanche and non-avalanche situations. An alternative point of view is to consider precision and recall rather than true and false positive rates (Figure 4b). The maximal precision that can be reached with our model is around 30% but with a very low value of recall

(below 5%). With higher values of recall, the precision ranges between 2 and 10%.

### 3.3   Variable importance

As described in Section 2.6.4, the predictive power of the input variables can be estimated in two ways.

First, we computed the feature importance of all variables and aggregated (summed) them by groups, as defined in Table 1 (Figure 5). The most important variables are related to snow depth (Figure 5) and, in particular, the new snow amounts or

snow depth variations. Variables related to dry snow stability appear to also be of large importance (13.6%) but with much more variables in the corresponding group: 25 dry stability indices, whereas there are only four variables in the new snow

**Table 4.** Confusion matrix for the evaluation dataset: observed and predicted avalanche situations ("Avalanche") and non-avalanche ones ("Non-avalanche") summed over elevation and aspect ranges. A threshold value of 0.01 is used, i.e., predicted probabilities over 0.01 are considered to identify avalanche situations. The corresponding recall is 75.3%, the false positive rate is 23.6% and the precision is 3.3%.

|  |  | Predicted | |
|  |  | Avalanche | Non-avalanche |
| --- | --- | --- | --- |
| Observed | Avalanche | 1 895 | 623 |
|  | Non-avalanche | 55 005 | 178 357 |

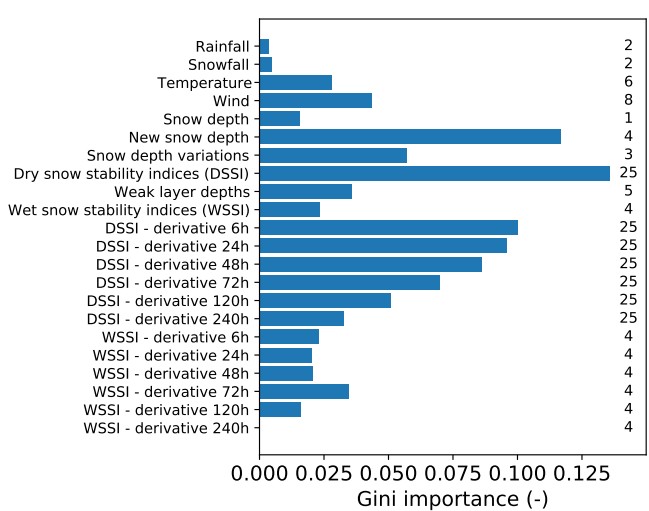

**Figure 5.** Feature importance (Gini importance) on train dataset, aggregated (summed) by groups of variables. The number of variables in each group is reported on the right.

depth group. The depths of weak layers is also of importance (3.6%). Derivatives of dry snow indices decrease in importance with time step, whereas for wet snow indices, the importance is more pronounced for a time step of 72 h. Temperature and wind are also important, even described with few involved variables. By contrast, snowfall and rainfall (on 24 h) are variables with low importance. The variability between years is limited (not shown), giving confidence in the robustness of these results. However, absolute values have to be taken with care as this analysis method is strictly valid only when the different variables are independent, which is far from the case we have here. We thus provide this analysis to check the main results according to previous knowledge and because of the popularity of this method, but the detailed results are of limited interest due to the presented limitations of this method in our study case.

Second, we studied the importance of variable groups by removing the data related to different groups of variables before learning and observing changes in evaluation scores. Specifically, we selected six subsets of the presented variables (see Table 1): the meteorological variables only (Meteo), bulk variables only (Simple snow), stability variables without derivatives

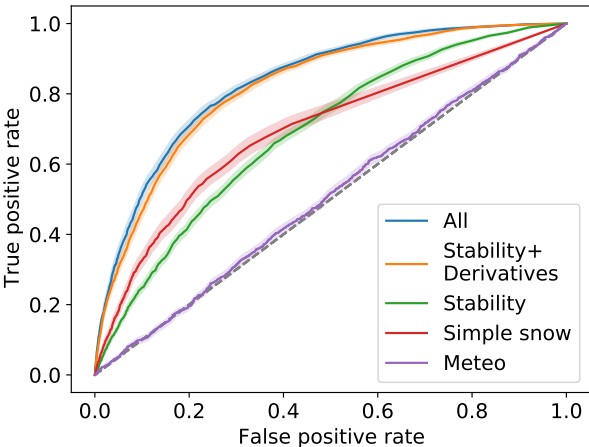

**Figure 6.** ROC curves of the model trained with different sets of variables. Shading represents the uncertainty by bootstrap on evaluation years (see methods section). Labels of subsets of variables correspond to those of Table 1. Scores associated to the optimal points (nearest to (0, 1)) are reported in Table 5

(Stability), stability variables and derivatives (Stability+Derivatives) and all variables (All). The ROC curves for all these subsets are presented in Figure 6. The associated scores for the optimal threshold are reported in Table 5. These thresholds are coherent with the base rate of our dataset. The ROC curve of the model trained only on meteorological variables is very close to the first bisector (Area Under the Curve AUC=0.09, Figure 6). In other words, this model is almost not much better than a random classifier. Using the simple snow variables (snow depth and new snow depth) allows for a first improvement in scores with an AUC of 0.19. Using the stability variables also allows for an AUC of 0.19 and combining it with the associated 174 time derivatives increases the AUC to 0.32. This result highlights the importance of time dimension in avalanche activity forecasting. The AUC of 0.32 for stability and derivatives is close to the value (AUC=0.33) obtained by using all variables. Moreover, the uncertainty linked to inter-annual variability is larger than the difference between the two latter approaches. This means that using all stability indices and their derivatives contains all relevant information available (in the context of the variables tested in this study) for discriminating avalanche and non-avalanche situations. The other scores (false positive rate FPR, recall, precision, see Table 3) present similar trends between groups compared to AUC. Some differences are nevertheless observed, with for instance a higher recall but a higher FPR for stability and derivatives compared to all variables, which highlights that the selection of an optimal classifier is always a question of compromise between these two scores.

**Table 5.** Predictive performance of the model trained with different sets of variables. The scores include area under ROC curve (AUC), false positive rate (FPR), recall and precision. We also report the associated optimal threshold used to compute these scores (associated to the point of the ROC curve nearest to the optimal one). Subsets of variables correspond to those of Table 1

| Subset | AUC | FPR (%) | recall (%) | precision (%) | threshold |
|---|---|---|---|---|---|
| Meteo | 0.009 | 49.9 | 51.7 | 1.1 | 0.025 |
| Simple snow | 0.195 | 32.9 | 65.3 | 2.1 | 0.001 |
| Stability | 0.188 | 38.1 | 65.9 | 1.8 | 0.01 |
| Stability and derivatives | 0.321 | 26.5 | 76.7 | 3.0 | 0.01 |
| All | 0.334 | 23.6 | 75.3 | 3.3 | 0.01 |

## 4   Discussion

### 4.1   Machine learning for predicting avalanche activity

The model performance in the studied area decomposed into eight aspects and three elevation bands, is summarized with the confusion matrix shown in Table 4. Values of recall (75.3%), false positive rate (23.6%) or precision (3.3%) may seem quite low compared to current literature. Hendrikx et al. (2005) or Kronholm et al. (2006) obtained accuracy for separation around 85% with regression trees and meteorological variables or simple snow variables (snow depth or simple melting model). The accuracy of our model is 76.5% but this metric may not be the most informative when classes are highly unbalanced, as in our problem because it mainly gathers information on non-avalanche situations. Sielenou et al. (2021) reported scores above 95% for accuracy but did not exploit other metrics. Hendrikx et al. (2014) reported a recall (focusing on observed avalanche situations) of 76 to 79%, close to our value of 75.3%. Some studies, such as Pérez-Guillén et al. (2022) or Mayer et al. (2022), did similar work using different targets (manually predicted avalanche hazard or measured stability) with accuracy also in comparable ranges (72 to 88%). Precision is highly influenced by the base rate (proportion of avalanche situations). Here, avalanche and non-avalanche situations are highly unbalanced. We nevertheless consider that the balance is representative of avalanche activity in the considered area. Moreover, low values of precision (around 3% for our model) are not uncommon for such difficult problems in related but different contexts (e.g. Rubin et al., 2012). Eventually, to compare our results to some studies with balanced dataset, the balanced precision should be considered, which is 76.1%.

However, it remains difficult to compare scores to other studies due to differences in evaluation methods and reported scores. All studies used different methods for defining a training and an evaluation dataset. In this study, we used a robust and conservative method, consisting in isolating winter seasons for evaluation. Indeed, with the snow melting between seasons, we get rid of the snowpack memory and provide a robust separation between training and evaluation datasets, leading to trustworthy evaluation results with our method. Moreover, we discard all the situations where the snow depth is less than

10 cm in the release zone, and situations outside the winter period where avalanche release is very unlikely. Consequently, our evaluation does not include the most obvious non-avalanche situations. It is thus more strict than Sielenou et al. (2021), for instance, who used the random forest out of bag method with oversampling of the minority class. It resembles the methodology of Hendrikx et al. (2014) who selected two independent years for evaluation. Our method may be used for future benchmarks to compare competing methods on a robust and homogeneous basis. In addition, the scores reported are not homogeneous between studies either. Some of them focus on global accuracy (e.g. Kronholm et al., 2006; Pérez-Guillén et al., 2022), others on accuracy per class (e.g. Sielenou et al., 2021) and a few propose other metrics such as recall, precision or F1 score (harmonic mean of precision and recall) (e.g. Hendrikx et al., 2014). The choice of the score depends on the goal of each study and must be adapted to it. However, limiting to a few values for summarizing the model performances limits the information available. These differences in the evaluation processes - both separation between evaluation and train sets and computed scores - limit the possibility of model comparison.

Our model predicts the probability that at least one avalanche occurs on a given day within a spatial unit corresponding to one elevation band (centred at 1800, 2400 and 3000 m) and one aspect (among 8 aspects). This spatial resolution enables to capture the spatial distribution of the expected avalanche activity in one region. This latter information is crucial to evaluate and to describe the avalanche danger at regional scale (Morin et al., 2020). This prediction goal is more demanding than a prediction at larger scales, as generally used in previous studies. For instance, if one avalanche occurs one day, it implies to identify that we have an avalanche situation but also in which aspect and elevation sector to be considered a success. An avalanche predicted in an other elevation or aspect will be considered as one false negative (in the elevation-aspect it really occurred) and one false positive (in the elevation-aspect it was predicted). It inevitably leads to lower performances for similar models but provides more precise information about the spatial distribution of the avalanche hazard (Statham et al., 2018). Indeed most studies considered avalanche activity at the scale of mountain ranges, of some thousands of $km^2$ (e.g. Kronholm et al., 2006; Hendrikx et al., 2014; Sielenou et al., 2021; Pérez-Guillén et al., 2022). These approaches have the advantage of using machine learning to also aggregate information at larger scales but provide a less geographically precise indicator of avalanche activity. More local approaches have the advantage of providing a relation between snow and meteorological conditions and observed or expected avalanche activity whereas aggregated approaches are closer to the final hazard assessment scale.

## 4.2 Added value of physical modelling of snow cover, stability analysis and time derivatives for predicting avalanche activity

We tested different input variables to train our model: meteorological variables, simple snow variables (mainly snow depth), stability indices and derivatives. We evaluated the added value of the different groups of variables with two different methods (described in Section 2.6.4). Meteorological information only was insufficient to predict avalanche activity with our method (Figure 6). Contrarily to many other studies (e.g. Buser, 1989; Mayer et al., 2022), we did not use observed meteorological information but large-scale modelled information (Durand et al., 2009). Thus, the meteorological information is uncertain and nearly identical for all aspects and elevations, while underlying snowpacks are generally significantly different. Therefore, we did not expect a good prediction at high spatio-temporal resolution with only meteorological information.

Most of the developed models used, at least, some basic output of snow cover models or observed snow evolution such as snow depth (e.g. Hendrikx et al., 2014). In our study, snow depth and new snow depth appeared as an essential variable in both methods used to estimate variable importance: its Gini importance is high (Figure 5) and adding it to the input variables improves a lot the model performance (Figure 6). This result is consistent with current literature identifying snow depth as the first statistical predictor for avalanche activity (e.g. Schweizer et al., 2003; Castebrunet et al., 2012; Sielenou et al., 2021). Some studies used more advanced diagnostics from snow cover models (e.g. Gassner and Brabec, 2002; Pérez-Guillén et al., 2022; Mayer et al., 2022) or computed expert aggregated variables similarly to what snow cover models do from temperature and precipitations (e.g. Kronholm et al., 2006). Snow modelling with physical models for taking into account snowpack history thus appears of high interest for automatic avalanche activity prediction.

The novelty of our model is to add a wide range of stability indices to reduce the complex information of snow cover models with the help of knowledge of physical processes and combine it with a time-dependent analysis with the use of time derivatives of stability indices. The combination of stability indices and derivatives is crucial in our random forest model (Figure 5). The time dimension has been identified as a critical information. Since the first statistical forecasts, differences between time steps, for instance on temperature (e.g. Obled and Good, 1980; Navarre et al., 1987), have been used. Conway and Wilbour (1999) also have developed a stability index that explicitly uses time derivatives. We here show that the use of time derivatives, especially in a statistical system that is not able to treat simultaneously different time steps, allows for an improvement of the prediction of avalanche activity. More generally, we showed that the introduction of stability indices and time derivatives could help identify avalanche-prone situations with machine learning models. This group of variables also gathers a great deal of information as it nearly replaces the information from other variables. Indeed, our results are quite similar when using only stability indices and their derivatives versus using all variables (Figure 6). This result indicates that stability indices combined with time derivatives are a relevant way to summarize the information of meteorological and snow cover models for the prediction of avalanche-prone conditions, which is a new way of validating the interest of such stability indices.

Computing feature importance can drive the selection of relevant input variables but correlations between variables can affect the computed importance. Re-training the full model with a subset of input variables provides a robust estimation of their effective added value. In particular, the analysis of feature importance allows for selecting the right time steps for derivative computations in the wide range of possibilities included (last column of Table 1). The most important derivatives are the short-time ones (6 to 72 h) for dry snow and 72 h for wet snow (Figure 5). This result is consistent with the knowledge of involved processes (van Herwijnen et al., 2018), whereas it was never demonstrated so far with a statistical approach. The spontaneous release in dry snow occurs during or immediately after snowfall whereas wet snow problems are more linked to the progressive wetting of the snowpack due to solar radiations (time scales of one to several days) or rain (e.g. Reuter et al., 2022). Variable importance allows for selecting the most relevant variables which may be kept for further work, especially on stability variables and derivatives, which our results prove to be of interest (Figure 6).

## 4.3 Other advantages and disadvantages of our approach

We used the EPA as the ground truth of avalanche activity. This dataset is unique in its spatial and temporal extension but mainly focuses on large avalanches often reaching valley floors. In consequence, the high-elevation avalanche activity and smaller avalanches are not reported, which leads to a limited number of avalanche situations in the dataset. Yet, for the spatio-temporal domain selected in this study (Haute-Maurienne, 1960-2018), the number of avalanche events reported in the EPA remains large enough (2518 avalanche situations). The local topography with steep slopes and the repartition of the recorded avalanche paths allow for a reasonable screenshot of the avalanche activity. However, the scarcity of reported avalanche events might become a problem in other regions as our balancing methods may become insufficient. Observation may not be possible every day (e.g., poor visibility or remote sites), and only avalanches are reported (i.e., no information on the observation that no avalanche occurred). This means that the dataset does not allow to clearly define non-avalanche situations: some situations may be identified as non-avalanche situations, while an avalanche occurred but was not reported. The data also suffers from uncertainty on the dates of avalanches. This may reduce the obtained score. Other data sources may be used to complement the avalanche observation dataset, such as observations from ski resorts (e.g. Giard et al., 2018) or satellite avalanche detection (e.g. Karas et al., 2022), but no other data source has the temporal extension of EPA, except archival data that require in-depth investigations which cannot always be undertaken (Giacona et al., 2017, 2021).

Moreover, we here trained the model with the Haute-Maurienne data. Some climatological or terrain features may lead to a predicted avalanche activity specific to the Haute-Maurienne area, especially with a higher sensitivity of certain aspects or elevations (e.g., during easterly returns). Hence, the model may not be transferable directly to other areas without a new calibration. Finally, this study presents a binary classification as there is rarely more than one avalanche per day and spatial unit (aspect-elevation), which limits the definition of several classes of avalanche activity. In the future, such machine learning techniques may benefit from the use of other sources of data to complement EPA data and identify more avalanches, such as remote sensing (e.g. Karas et al., 2022), infrasound (Mayer et al., 2020) or seismic detection (van Herwijnen and Schweizer, 2011).

In this study, we chose to treat all avalanche types in a single learning process, including dry and wet avalanches. Some previous studies separated different avalanche activities on pre-defined time periods (e.g. Obled and Good, 1980), or by type of avalanches, restricting to dry or wet avalanches (e.g. Mayer et al., 2022; Pérez-Guillén et al., 2022). If we assume that decision trees (or here, Random Forest) can capture dry avalanche activity on one hand and wet avalanche activity on the other hand, and if we provide information to discriminate between situations, such as liquid water content or height of wet snow in this study, then a decision tree (or an ensemble of them) will be able to be optimized on the overall avalanche activity by introducing a split in the overall tree to distinguish between dry and wet situations, if relevant. Some other studies also mix dry and wet avalanches, such as the MEPRA French operational avalanche activity indicator (Giraud et al., 2002). Moreover, the observation dataset does not always allow to infer the processes that led to the avalanche and some situations may remain uncertain in case of a mix of dry and wet snow in the snowpack. For the forecasters, complementary information may be

provided with additional tools to identify the processes or situations involved, such as the avalanche problem types suggested by Reuter et al. (2022).

The impact of using physically-based indices of snow stability as predictors of avalanche activity instead of simpler variables was studied through a specific statistical tool, namely random forests. This method is popular due to its simple background (decision trees (Breiman et al., 1984)) which allows for in-depth analysis and interpretation to some extent and its capacity to represent non-linear phenomena (e.g. Sielenou et al., 2021; Pérez-Guillén et al., 2022; Mayer et al., 2022). Many other statistical methods are available but random forests have been shown to be as relevant as other ones (e.g. Sielenou et al., 2021).

We introduced time derivatives and cumulative values to represent the importance of history for snowpack-related processes. Methods in the range of recurrent neural networks are specifically designed to cope with processes having a memory of previous states (Hochreiter and Schmidhuber, 1997). These alternative statistical methods could be further compared to our random forest approach. It may provide improvements in the prediction scores or strengthen our results on the effectiveness of combining snow physics and machine learning for predicting avalanche activity.

Our results were obtained with a reanalysis of meteorological and snow conditions, that is to say, input data that have been retroactively corrected with all available observations. This may not be completely representative of operational forecasting (prediction in the future) situation in which models are corrected by observations of the past but run unconstrained for the forecast. This transposition to the forecasting context would be the next step in terms of complexity for machine learning methods. However, the use of the reanalysis allows for a better evaluation of the capabilities of the machine learning model 460 with fewer input errors, which was the goal of this paper.

## 5   Conclusion and outlooks

This paper combines snow cover modelling, mechanical stability indices and observational data through machine learning for avalanche activity prediction. In particular, we considered numerous stability indices and their time derivatives. To evaluate the random forest model, we defined a robust method adapted to the specific behaviour of the snowpack (long-term memory). 465 This evaluation was conducted on three district municipalities of the French Alps with 58 years of a comprehensive dataset of avalanche observations, with a high spatial resolution (8 aspects and 3 elevation ranges) and an extended set of variables describing both meteorological, mechanical stability variables and their time evolution.

    The combination of snow physics through snowpack modelling, stability indices and their derivatives, and random forest proves to be useful for avalanche activity prediction. The snow depth and new snow depth remain the most important predictors 470 but this study highlights the interest in using mechanical stability indices and their derivatives. This is the primary finding of our research as this had never been demonstrated with such a large variety of indices and their derivatives in previous studies (e.g. Zeidler and Jamieson, 2004; Kronholm et al., 2006; Hendrikx et al., 2014; Sielenou et al., 2021), even the rare ones using simple stability indices within machine learning models (Mayer et al., 2022). Our results also underline the interest of physically-based snow cover models and stability indices for identifying avalanche-prone conditions.

Obtained scores of recall (75.3%), false positive rate (23.6%) and precision (3.3%) are consistent with current literature with similar goals and methods. These scores illustrate the difficulty to predict avalanche occurrence with high spatio-temporal resolution, even with the data and modelling tools currently available. Moreover, we used a rather strict evaluation method leading to lower but robust and conservative scores, which are not directly comparable to other studies (e.g. Sielenou et al., 2021). Hence, this method may be seen as the first step for future formal comparison between approaches. More widely, with its high spatio-temporal resolution and use of physical and mechanical models, our study opens the perspective to improve modelling tools supporting operational avalanche forecasting.

We here focus on the avalanche activity reported by EPA. The method may be extended in the future to other target variables describing more precisely avalanche hazard such as release volumes or typical situations (Schweizer et al., 2003; Statham et al., 2018; Reuter et al., 2022; Mayer et al., 2022). Similarly, we used meteorological reanalysis for snow modelling for the quality of the data but this may not be completely representative of forecast conditions and tests have to be conducted with re-forecasts rather than reanalysis.

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

*Author contributions.* Léo Viallon-Galinier developed the model code and performed the simulations. Léo Viallon-Galinier prepared the manuscript with contributions from all co-authors.

*Competing interests.* The authors declare no competing interests.

*Acknowledgements.* IGE/INRAE and CNRM/CEN are members of LabEx OSUG. Authors are grateful to the numerous people from ONF-RTM that contributed to the EPA data collection.