# Peer review of "Does combining modelled snowpack stability with machine learning help with predicting avalanche activity?"

_The Cryosphere, 2022_

## Referee Comment (RC2)

[referee-annotated manuscript omitted]

---

## Author Comment (AC1)

**Answer to Frank Techel (RC1)**

Léo Viallon-Galinier        Pascal Hagenmuller        Nicolas Eckert

> The authors present a random-forest algorithm, which predicts the occurrence of natural avalanches running to the valley bottom in the Haute-Maurienne part of the French Alps. The algorithm is trained using a long-term record of avalanche observations, a highly unbalanced data set with 100 times more non-avalanche days compared to avalanche days. From my perspective, the novel - and certainly very challenging aspect of this study, is the prediction of (often single) avalanche events for aspect-elevation segments. The algorithm's predictive performance is characterized by recognizing many of the observed avalanche days, but having a very high false-alarm rate (only 3% of the predicted avalanche days coincided with observed avalanche days). The manuscript is well written, and most sections are easy to follow. Questions, however, arise with regard to the definition of the target variable (Sections 2.1-2.3, 2.5.1, Discussion), the stability indices for dry snow (Sect. 2.4.1), and the way the variable importance is presented and interpreted (Sect. 3.2 and Fig. 4).

> Please find below some comments regarding these three points. I hope these comments will be helpful in improving the manuscript.

We thank Frank Techel for this detailed review that will improve gloabally the manuscript. We provide below a point by point answer to all his comments.

**General comments**

**(1) Definition of the target variable and subset used for training and testing**

> - You defined avalanche days (AvD) and non-avalanche days (nAvD) by aspect-elevation-segment (AE segment). For a specific AE segment, an AvD is fulfilled if at least one avalanche running to the valley bottom (below the blue line in Figure 1) was observed, while nAvD are all other days (l 148-149). If possible, please provide an indication regarding the minimal avalanche size that would be typically required to reach this run-out zone in the study area.

The observation network was designed at the end of the 19th century when avalanche sizes were not yet normalized. Hence, the european avalanche size scale is not explicitly used. The minimal size is indirectly defined for each avalanche path by the position of the observation threshold that should be crossed by avalanches to be recorded by the observer. Empirically, we can imagine that no size 1 avalanche are recorded. Avalanches of size 2 may be recorded, especially if an accident is related to this avalanche or if the avalanche reached high altitude infrastructure and most of the recorded avalanches may be of size 3 or more. However, this was never explicitly evaluated. We thus prefer not to give an indication that will not be properly supported. Moreover, due to the high number of avalanche paths in Haute-Maurienne and the steepness of the slopes, we believe that EPA provides a excellent overview of local natural avalanche activity, as we point out in the description of the dataset: *Besides, the steep topography of Haute-Maurienne reduces the effect of the threshold of observation as most of the avalanches reach the valley floor, providing a representative screenshot of the overall avalanche activity in the area.*

> - Overall, I think that the description of AvD and nAvD could be improved. Particularly, what is considered a nAvD is not fully clear. Furthermore, as nAvD were 100 times more frequent compared to AvD, it could be valuable to use a more strict definition of nAvD, excluding for instance days when avalanche activity was uncertain (l 96-101). Not doing so, will inevitably reduce the performance statistics, not because the model performs poorly, but because the target variable is uncertain.

We will rewrite paragraph 2.5.1 to make clearer the difference between avalanche and non-avalanche situations: *Avalanche activity is based on EPA records in the selected area. Days were classified into two categories: avalanche*

*and non-avalanche situations. For a given day, aspect sector and elevation band, it is considered as an avalanche situation if at least one avalanche is reported for this day, aspect and elevation band and a non-avalanche situation if no avalanche observation were reported..* Moreover, we change the name from avalanche/non-avalanche day to avalanche/non-avalanche situation as a situation is defined for a given day and AE sector.

It is not obvious how to chose non avalanche days to remove. When the uncertainty is provided, it concerns the date of avalanche days. It is not possible to remove all days in the uncertainty ranges as these uncertainties can be important, up to several months for most remote sites that are not observed systematically. Moreover, other uncertainties exists but are not reported. When no information is reported, we assume that no avalanches occurred. However, it can simply signify that the visibility was too low to observe anything or the observer was not available a given day. In these cases, no specific information is reported to discriminate between non-avalanche days and uncertain situation. This is one of the major drawbacks of this dataset. We currently work on alternative datasets to overcome these limitations, but no other dataset allow for a constant observation method on such a long period. We will improve the discussion on this limitation in the discussion section.

- Some avalanche events had uncertain dating (l 96). Please indicate the number of these events.

An uncertainty is associated to each event. We will provide indication on the part of the recorded events that have uncertainty over 1 and 3 days in the revised version: it concerns respectively 28.6 and 23.6% of the dataset on the considered area.

- You removed avalanche events with an uncertainty on the release date of more than three days from the data set (l 97-98). Were these days and AE segments then treated as nAvD, or removed from the data set?

Observations are not considered when uncertainty is more than three days. The 23.6% of observations that have an uncertainty over 3 days, most of them concerns remote site that are not visited regularly and 16.5% of observations thus have an uncertainty of one month or more. It is not possible to remove complete months from the analysis. For each day and AE segment, the status is then determined depending on whether an avalanche is reported or not in the remaining data. This remark will be taken into account in the rewriting of paragraph 2.5.1.

- In case the uncertainty of the release date was two or three days, you assigned the last day as the date of release (l 98-99). Did you treat the two previous days as nAvD, or were these removed from the data set? On l 146-148 you explain why the time derivatives are required and that avalanches may release when the stability is lowest. This is somewhat different to how you assigned the avalanche release date when this was uncertain.

Previous days are assigned depending on other observations. It is considered an avalanche day if an observation related to the considered day and AE segment is reported and non avalanche day otherwise. Information on previous day may help the model to deal with the uncertainty on dates but we fully agree that the inclusion of derivatives is to discriminate the most critical moment is something unrelated.

- You state that the data set provides a "nearly exhaustive screenshot of natural avalanche activity" (l 93). To me, less than 3000 avalanches in 110 paths in 58 years do not seem exhaustive at all. Consider rephrasing this sentence, for instance to "a representative screenshot of avalanche activity of avalanches running to valley floor" or similar.

We will rephrase the sentence to *provide a representative screenshot of the overall avalanche activity in the area*. The word *exhaustive* is indeed incorrect, the dataset do not report exhaustively avalanches, but we believe that due to the specific context of Haute-Maurienne (steep slopes, a lot of avalanche paths recorded), the dataset is an excellent indicator of the overall avalanche activity.

- There are 110 avalanche paths and 24 AE segments. - If you consider the topographical distribution of potential start zones, are all AE segments equally often represented? For instance, the distribution in Figure 2 shows that there were 100 times more avalanches in the South aspects compared to the North-East aspects. Is this due to more start zones in South aspects or because activity was indeed higher? Providing more information on the distribution of start zones per AE segment would help the reader to understand this relationship. Consider showing the AE distribution of potential start zones in the study area, maybe in a plot similar to Figure 2. If they were distributed rather unequally, please discuss how you considered this in the analysis, and what impact this may have on the results.

Avalanches are observed only if they are located in pre-defined avalanche paths and reach the observation line. However, information reported contain the elevation of departure of the avalanche and the aspect of the area where it started. This may not be directly linked to avalanche path information as a path may be globally south-oriented but sides may look south-east or south-west, or even East or West for avalanche paths with large departure zones. We use the information related to the precise recorded avalanche as soon as it is available (most of the time), that is why we presented the data for the observed avalanches rather than for the avalanche paths.

Moreover, as the valley turn, we have globally north and south facing paths in Lanslevillard and more South-East and West facing path in Bessans for instance. However, even though a wide variety of aspects is represented in terms of avalanche paths, this does not ensure that all paths are equivalent. Some may have more forest than other, or different vegetation influencing susceptibility to avalanches, some may be steeper than others, etc. We cannot ensure an equal representation of all possible aspects and elevations with similar conditions.

However, as we point out in the description of the dataset, we believe that due to the high number of observed avalanche paths and the steepness of the slopes, the recorded EPA avalanche activity is a good proxy of overall avalanche activity of the Haute-Maurienne valley. Hence, when the goal is to predict the avalanche activity of Haute-Maurienne, the use of a realistic avalanche activity, including unbalance between elevation and aspects seem relevant. This means that the model may not be directly transferable to other areas. We will introduce this in the discussion: *we here train the model with the Haute-Maurienne data. Some climatological or terrain features may lead to a predicted avalanche activity specific to the Haute-Maurienne area, especially with a higher senitivity of certain aspects or elevations (eastern crests during easternly returns). The model may not be transferable directly to other areas without a new calibration.*

- You attempt to predict both dry-snow and wet-snow avalanches with the same algorithm. I suspect that this probably contributes to the poor performance of the algorithm as a dry-snow avalanche can't be correctly predicted by a tree, which learned conditions favorable for a wet-snow avalanche, and vice versa. This should be discussed.
- Does the EPA provide information on the wetness of the avalanche? Please briefly indicate whether it did or not and if it did, why you preferred to develop one rather than two algorithms. It could also be discussed that splitting the data into wet and dry snow conditions using the simulated stratigraphy and learning two separate algorithms may have helped to address the different release mechanisms in a more appropriate manner, which would potentially also cause fewer false alarms.

The EPA being an observation of avalanche deposits, with remote observations from valleys, it provides few information on processes in starting zone. In particular, although the deposit is often described as mainly dry or mainly humid, the wetness of the snowpack in the starting zone is not reported. It is therefore difficult to classify between dry and wet avalanches based on information reported in the dataset. More generally, classification between dry and wet avalanches is not always obvious, especially during the progressive wetting of the snowpack (from top to bottom) during spring or when dry snow falls over a wet snowpack.

By removing litigious situations and using snow cover modelling, it remains possible to define two subsets of wet and dry snow. However, we do not agree on the need of splitting a priori the two types of avalanche processes. We use the tree-based RF model. If the wetness of the snowpack is a critical factor to identify the situation, it should be selected during the optimization process as one of the top split in the tree directly by the model, especially as we provide relevant indicators to identify if the snowpack is rather dry or wet, such as the height of wet snow or the mean liquid water content. Then, the two branches will analyze different characteristics depending on the situation (dry or wet). The model should then be able to deal with different situations. In the dry snow this is also required as we have to identify situations where a persistent weak layer is involved from situations where only the new snow have to be considered, for instance. We thus do not think that a split between dry and wet situations would help the classification, even though we know that it is a common approach in the avalanche community.

We nevertheless tested to focus on the wet snow situations, as it is closer to the analysis done by forecasters. We extracted the situations for which the snowpack is mainly wet from the whole dataset (both for non avalanche days and avalanche days), based on snow cover modelling. The performance on the resulting model, focused on wet snow was not better than the full model. We then do not pursue in this direction.

We will introduce a paragraph in the discussion section to summarize this.

- Why did you pick 15 Oct until 15 Mar as the winter season? 15 Oct seems rather early, and 15 Mar rather late. Please explain.

The period is from 15 Oct to 15 **May**. We will correct this error in our original version in the revised manuscript. The choice of dates inevitably contains arbitrariness. We wanted to include a large variety of situations. In France, the avalanche bulletin is produced from early November to early June, which is coherent with the selected date range. Our choice is also coherent with the choice of other studies, such as [Sielenou et al., 2021], for instance. We will specify these reasons in the text.

- Why did you use a 1 cm threshold as minimal snow depth? (l186) Or did you use 10 cm, as stated later in the manuscript (l 299)? Both values seem rather low snow depth values considering that avalanches must be rather large to reach the run-out zones. Also along this line: how did you treat cases when there was no snow in a lower elevation band, but some snow in the highest elevation band. I suspect that avalanches running almost to the valley bottom are probably rather unlikely in these situations (-> nAvD), even if conditions in the start zone would favor avalanche release.

Sorry for the inconsistency. We will correct the value line 186. We use a threshold of 10 cm to remove days with no or few snow on ground. The threshold is inevitably arbitrary. We would like to keep all situations that could lead to avalanches and therefore select a conservative value. It therefore allows for a significant undersampling, especially on low elevation bands. The statistical algorithm then have the role of selecting optimal values to separate between avalanche and non avalanche situations. Hence, this threshold is chosen to be conservative and not optimal in any way.

For the way we compute avalanche and non avalanche day, it is important to notice that there is no relation between the three elevation bands and eight aspect sectors we consider. It provides 24 situations composed of meteorological and snow conditions as well as avalanche observations each day. We propose to rename avalanche day and non avalanche day to avalanche and non-avalanche situation and better explain this specific approach in the material and method parts to limit misunderstandings.

**(2) Presentation and interpretation of variable importance (Sect. 3.2 and Fig. 4)**

- Fig. 4 shows the variable importance, aggregated (summed) by groups of variables. This is a rather unusual way of presenting variable importance and makes the interpretation of the plot rather difficult. For instance, snow depth and variations (SDV) and dry snow stability indices (DSSI) have the same cumulative Gini importance (about 0.18), but the first contains 7 variables, the latter 30. This means that on average each SDV variable has a higher importance ($0.18/7 = 0.025$) compared to a single DSSI variable ($0.18/30 = 0.006$). This only becomes clear from the plot when making these calculations. This is also somewhat indicated in the text (l 259-260).

Considering individual importance is misleading because the variables we use have important redundancy. The importance is therefore shared between several variables containing redundant information. We propose this visualisation because it is a common approach and allow to test easily selections of variables. However, if the results indicate main trends, precise values have to be handled with care as we point out lines 235 to 238. We propose to add a reminder that absolute values have to be treated cautiously in the result section.

- To me, it was not intuitive, which of the 7 variables belong to snow depth and variations (SDV). I was able to figure this out after going back to Table 1. Maybe you could somewhere describe this more clearly in Table 1 and/or Figure 4? For the other variable groups, this was clear.

We will precise it both in Table 1 and legend of Figure 4.

- Did the depth of the weak layers, described in Table 1, not play a role in the RF models? It seems to be missing in Figure 4.

Depth has an importance, we forgot this group in Figure 4. We will add this data in the revised version.

**(3) Variable definition (Sect. 2.4.1)**

You selected the five weakest layers in each profile (l133-136). Please explain why you used five layers and not just the weakest one. Furthermore, I wonder whether the stability of the five weakest layers isn't highly correlated? What would happen if you train the RF only with the weakest layer? Please elaborate more on how you selected the five weak layers if the local minima for Sn, Sa, Sr, + two crack propagation indices were

> in five different layers, and how if they all indicated the same weak layer.

We have five ways of identifying a weak layer through the five dry stability indices we selected. That is why we selected five weak layers (see line 133). In some situations, the five weakest layers may be highly correlated, if no identical. In some other situations, we know they are different. Beyond this question on the weak layers, a lot of our variables are highly correlated.

**Technical comments**

Thanks for these detailed comments. We do not answer to all technical comments that will be taken into account as proposed.

> - l 60: consider rephrasing this sentence as machine learning approaches evaluation is somewhat awkward to read
> - l 63: consider replacing of of interest with suitable, or similar
> - l 72: in this study could probably be deleted
> - l 77: consider removing largely
> - l 87: consider adding was before extensively
> - Figure 1: please show the runout area more clearly, for instance by shading it

We tried both representation and find that a shading does not allow for a better interpretation of the figure.

> - l 97-98: consider rephrasing the second part of this sentence (from the data set at the end of the sentence)
> - l 144: typo Considering –> considering
> - l 146-148: somewhat awkward to read, consider splitting or rephrasing this sentence
> - l 180: consider rephrasing the beginning of this sentence to We use two classes or similar
> - l 186: You mention that the first selection criteria causes undersampling. What impact did the second selection criteria have?

We will add "*which corresponds to an oversampling of the minority class*" at the end of the paragraph.

> - l 207: typo probabilityy –> probability
> - l 215: Consider changing truly to correctly, or similar
> - l 243: typo closed –> close
> - l 250: add day after avalanche
> - l 298: what does leading to strong results mean. A recall of 3% is not really strong. Consider rephrasing.

We propose to rephrase to *leading to trustworthy evaluation results*.

> - Discussion: It would be rather nice to see an exemplary time series of the model predictions for one winter season for all 24 AE segments, together with the corresponding observed avalanche activity. This may help the reader to get a better impression on the correlation between avalanche activity and model predictions.

We think that 24 different AE segments would not bring relevant information while overloading the paper. The main question of the paper is the interest of stability indices in combination with machine learning algorithms. We hence propose to include an illustrative example on one year, chosen to be representative of the results of the model.

> - l 351-353: this statement is correct, but maybe more importantly, this lowers the observed performance of the classifier as AvD predictions may be counted as a false alarm when in fact there was a (smaller) avalanche.

We agree with this remark and will include it in the revised version of the paper.

---

## Author Comment (AC2)

**Answer to Karl W. Birkeland (RC2)**

Léo Viallon-Galinier          Pascal Hagenmuller          Nicolas Eckert

**General comments**

> In this paper the authors present a method using random forests to predict natural avalanches running to the valley bottom in the French Alps. Their methods appear to be solid, and the question they are trying to answer is important. In comparison to previous research, the novelty of their approach is that they make their predictions at the spatial scale of specific elevations and aspects. The paper is generally well-written and clear. I believe this research makes a valuable contribution, but I also feel there are issues that should be addressed prior to publication.

We thank K. W. Birkeland for this detailed and useful review. We answer point by point to the different issues raised below.

> Here are a few of the major issues that I believe should be addressed:
>
> - It would be helpful for the reader to better understand the spatial characteristics of the starting zones of the approximately 110 avalanche paths in the study area. Looking at Figure 1, it appears that most of the starting zones will have either a NW or a SE aspect. I am not sure about the distribution of the starting zone elevations. A Figure like Figure 2 (which shows the distribution of avalanche events by aspect and elevation) should be created for the avalanche path characteristics. In fact, it would be useful to pair this new Figure with Figure 2 so the reader could assess the effect of the avalanche path characteristics on the number of avalanches in each elevation/aspect zone.
> - Along these same lines and again looking at Figure 1, I assume that the elevations and aspects of the avalanche starting zones are not evenly distributed in the 24 classes (three elevation and eight aspect categories). How does this affect the analyses? I understand that the authors would like to use the 24 elevation/aspect categories used in avalanche forecasts, but I wonder if it is appropriate to use all 24 categories for a dataset that appears to be unbalanced in the distribution of avalanche starting zone characteristics? How is this affecting their results?

Avalanches are observed only if they are located in pre-defined avalanche paths and reach the observation line. However, information reported contain the elevation of departure of the avalanche and the aspect of the area where it started. This may not be directly linked to avalanche path information as a path may be globally south-oriented but sides may look south-east or south-west, or even East or West for avalanche paths with large departure zones. We use the information related to the precise recorded avalanche as soon as it is available (most of the time), that is why we presented the data for the observed avalanches rather than for the avalanche paths.

Moreover, as the valley turn, we have globally north and south facing paths in Lanslevillard and more South-East and West facing path in Bessans for instance. However, even though a wide variety of aspects is represented in terms of avalanche paths, this does not ensure that all paths are equivalent. Some may have more forest than other, or different vegetation influencing susceptibility to avalanches, some may be steeper than others, etc. We cannot ensure an equal representation of all possible aspects and elevations with similar conditions.

However, as we point out in the description of the dataset, we believe that due to the high number of observed avalanche paths and the steepness of the slopes, the recorded EPA avalanche activity is a good proxy of overall avalanche activity of the Haute-Maurienne valley. Hence, when the goal is to predict the avalanche activity of Haute-Maurienne, the use of a realistic avalanche activity, including unbalance between elevation and aspects seem relevant. This means that the model may not be directly transferable to other areas. We will introduce this in the discussion: *we here train the model with the Haute-Maurienne data. Some climatological or terrain features may lead to a predicted avalanche activity specific to the Haute-Maurienne area, especially with a higher senitivity of*

*certain aspects or elevations (eastern crests during easterly returns). The model may not be transferable directly to other areas without a new calibration.*

> • Another issue is the inclusion of both dry and wet snow avalanches in the same analysis. This was also pointed out by the other reviewer. Since we know that the avalanche release mechanisms for these two primary categories of avalanches are quite different, as are the meteorological factors that lead to instability, why are these included in the same analysis? Perhaps this is because both wet snow stability indices and dry snow stability indices are included? Wouldn't it be better to split all the avalanches into "dry" and "wet" categories, and then proceed with the analysis on each of these two subsets of the data?

The EPA being an observation of avalanche deposits, with remote observations from valleys, it provides few information on processes in starting zone. In particular, although the deposit is often described as mainly dry or mainly humid, the wetness of the snowpack in the starting zone is not reported. It is therefore difficult to classify between dry and wet avalanches based on information reported in the dataset. More generally, classification between dry and wet avalanches is not always obvious, especially during the progressive wetting of the snowpack (from top to bottom) during spring or when dry snow falls over a wet snowpack.

By removing these litigious situations and using snow cover modelling, it remains possible to define two subsets of wet and dry snow. However, we do not agree on the need of splitting a priori the two types of avalanche processes. We use the tree-based RF model. If the wetness of the snowpack is a critical factor to identify the situation, it should be selected during the optimization process as one of the top split in the tree directly by the model, especially as we provide relevant indicators to identify if the snowpack is rather dry or wet, such as the height of wet snow or the mean liquid water content. Then, the two branches will analyze different characteristics depending on the situation (dry or wet). The model should then be able to deal with different situations. In the dry snow this is also required as we have to identify situations where a persistent weak layer is involved from situations where only the new snow have to be considered, for instance. We thus do not think that a split between dry and wet situations would help the classification, even though we know that it is a common approach in avalanche community.

We nevertheless tested to focus on the wet snow situations, as it is closer to the analysis done by forecasters. We extracted the situations for which the snowpack is mainly wet from the whole dataset (both for non avalanche days and avalanche days), based on snow cover modelling. The performance on the resulting model, focused on wet snow was not better than the full model. We then do not pursue in this path.

We will introduce a paragraph in the discussion section to summarize this.

> • The other reviewer also mentioned another issue I believe needs to be addressed. The dataset does not include all avalanches that occurred, but rather it consists predominantly of avalanches running to the valley floor. I assume these are almost all quite large avalanches. Can you provide a range of the size of the avalanches? Are they all Size 3 (on the Canadian or the U.S. destructive scale) or larger? Or perhaps size 4 or larger? What effect do the authors believe that this bias toward large avalanches has on their results?

The observation network was designed at the end of the 18th century when avalanche sizes were not yet normalized. Hence, the avalanche size is not explicitly used. The minimal size is indirectly defined for each avalanche path by the position of the observation threshold that should be crossed by avalanches to be recorded by the observer. Empirically, we can imagine that no size 1 avalanche are recorded. Avalanches of size 2 may be recorded, especially if an accident is related to this avalanche or if the avalanche reached high altitude infrastructure and most of the recorded avalanches may be of size 3 or more. However, this was never evaluated. We thus prefer not to give an indication that will not be properly supported. Moreover, due to the high number of avalanche paths in Haute-Maurienne and the steepness of the slopes, we believe that EPA provides a good overview of avalanche activity, as we point out in the description of the dataset: *Besides, the steep topography of Haute-Maurienne reduces the effect of the threshold of observation as most of the avalanches reach the valley floor, providing a representative screenshot of avalanche activity of avalanches reaching low altitudes.*. Then, we do not expect a bias on the results, as least as long as the model is not applied on other areas. We will develop this idea in the discussion.

> • While the authors reference some of the more recent work on predicting avalanches with random forests, I feel like they might want to also reference some early work that attempts to better predict avalanche activity using the statistical techniques available at that time. These older papers had more the more modest goal of trying to predict avalanche days (without elevation/aspect of the starting zones), but

they were a first step in this direction. This does not have to be a comprehensive review at all, but just a sentence or two with some references would be nice to see. Some older examples exist of researchers using discriminant analysis (examples: Bovis, 1977; Foehn and others, 1977), nearest neighbor techniques (example: Buser, 1983), and binary regression trees (example: Davis and others, 1992). Also, who was the first to use random forests for this type of work? Perhaps one of the authors who you already reference?

Thanks for pointing this lack. We will add a paragraph to the introduction to shortly summarize the history of machine learning and avalanches. The pioneering works were performed by Bois and Foehn in the 70s, with linear methods, while the first use of classification trees were by Davis et al in the late 1990s and random forest models were firstly used in the 2010s (e.g. Mitterer et al, 2013).

- Finally, one thing that perplexes me about this research is why new snowfall is rated so low in importance (Figure 4). This is completely different than prior research, which typically rated snowfall as the most important factor for dry avalanche release. Why do the authors believe this is the case? Is it because the "snow depth and variations" class is capturing this essential information? Or is it because of this information is captured (fully or partly) in some of the stability indices? Or is it the mixing of the dry and wet snow avalanches into one dataset? It might also be related to the fact that the dataset consists of only large avalanches. What do the authors think?

The variables we use contain a lot of redundancy and correlations. The random forest select the variable that allow the best separation into two groups at each step. We observe here that post-processed variables such as snow depth variations or new snow depth on 24, 72, 120h seem to be slightly more relevant than bulk snowfall the given day (or on 3 days), This is not contradictory with previous studies as it does not mean that snowfall do not contain relevant information. It just means that other variables are more relevant for the information related to new snow.

We have several possible ways of explanation. The first is that contrarily to most of the previous studies, we use large-scale modelled meteorological information rather than locally observed meteorology. We know that the Haute-Maurienne massif experience some heterogeneous meteorological conditions, especially during easterly return events, for which modelled meteorological information may not be fully representative. The second one is that most of the meteorological information we consider (precipitations and wind) are identical for all aspects and elevations and temperature are identical for all aspects and highly correlated between elevations whereas we know that the snowpack are generally quite different. The snowpack variables are able to summarize the history of past conditions that have built up the snowpack while meteorological information is not at the correct time scale for this. We will add a paragraph in the discussion to discuss this result on meteorological variables: *Meteorological information is not sufficient by itself. Contrarily to many other studies [e.g. Buser et al., 1989; Mayer et al., 2022], we do not use observed meteorological information but large-scale modelled information [Durand et al., 2009]. Thus, the meteorological information is uncertain and nearly identical for all aspects and elevations while underlying snowpack are generally significantly different. We then did not expect a good prediction at high spatio-temporal resolution with only meteorological information.*

Despite the above comments, I believe this is valuable research and is deserving of publication once the authors address or respond to these issues.

I have also attached an annotated PDF, which includes corrections to some typographical errors, as well as further suggestions and suggested wording changes.

We gathered our answers to the attached comments below.

I hope the authors find my comments and suggestions useful.

Karl Birkeland

Some possible older references (the authors may have other/different older references they wish to cite):

Bovis, M.J. 1977. Statistical forecasting of snow avalanches, San Juan Mountains, Southern Colorado, U.S.A. Journal of Glaciology 18(78), 87-99.

Buser, O. 1983. Avalanche forecast with the method of nearest neighbors: An interactive approach. Cold Regions Science and Technology 8, 155-163.

Davis, R.E., K. Elder, and E. Bouzaglou. 1992. Applications of classification tree methodology to avalanche data management and forecasting. Proceedings of the 1992 International Snow Science Workshop, Breckenridge, Colorado, 123-133 (available at: https://arc.lib.montana.edu/snow-science/item.php?id=1245).

Foehn, P.M.B. and others. 1977. Evaluation and comparison of statistical and conventional methods of forecasting avalanche hazard. Journal of Glaciology 18(78), 375-387.

**Attached comments**

We only detail hereafter the main comments of the PDF. We will take into account all the detailed suggestion in the attached PDF in the revised version.

> Page 1: I would suggest re-wording the title to make it more direct, while keeping the same meaning. I'm also not sure that "snow physics" is appropriate... perhaps it would be more accurate to state that you are really using modeled stability indices? Another thing that is important to emphasize throughout the paper is that we are talking about predicting large avalanches in this study (that go to the valley floor). Given all this, one suggestion would be: Does combining modeled stability indices with machine learning help with predicting large avalanches?

We propose to change the title to *Does combining modelled snowpack stability with machine learning help with predicting avalanche activity?*. We believe that the method presented here is not specific to large avalanches. Moreover, in the specific case of Haute-Maurienne, even though EPA observation dataset record avalanches that reach the valley floor, due to the specific geography of this area and the large number of observed paths, it is representative of the overall avalanche activity.

> Page 4: Are all the avalanches in the database naturally triggered? Or are there also some artificially triggered avalanches?

Avalanches reports are based on the observation of avalanche deposits. Observers have few information on the origin of the avalanches. The observation network was designed to give an overview of natural avalanche activity. Hence, natural avalanches are natural ones. However, we cannot ensure that no triggered avalanches are present in the dataset.

> Page 11: This is interesting. I am curious why snowfall does not look important in Figure 4, and rainfall also does not look important. It seems that snowfall should be among the most important variables, and the other snowfall variables (snow depth and variations in depth) are important. I will be interested to see how this is explained in the Discussion.

We answered in the general comments and will include further discussion on this point.

> Page 12: This is a surprising finding since new snow is often one of the most important variables for discriminating between avalanche and non-avalanche days.
>
> I find this to be really surprising since others in the past have had some reasonable results (though definitely not perfect) with only looking at meteo variables such as new snow, wind and temperatures.
>
> I am hoping you will discuss this in your discussion section.

We answered in the general comments and will include further discussion on this point.

> Page 14: I understand the other measures, but I think it would be helpful for the reader if you more explicitly explained "threshold" and what a value of that threshold means in the context of this Table. What is a "good" threshold, or is there such a thing? Larger numbers or smaller numbers or ??
>
> I read through some of the explanation in Section 2.6.3, but it was still not clear to me so I went and did my own work to try to better understand it.
>
> Now that I understand it better, perhaps your wording is OK. But, you could have another look and see. I would like it if there was something even in this figure legend that told us how to interpret these threshold values. Clearly they don't line up the same as the AUC values for the different sets of variables.

There is no good or bad threshold here, we reported this value as an information, as it traduces some details of the behavior of the model. We will adapt the legend to make it clear that, contrarily to other values, this is not a score, but an additional information.

---

## Author Comment (AC3)

**Answer to Simon Horton (RC3)**

Léo Viallon-Galinier        Pascal Hagenmuller        Nicolas Eckert

**General comments**

> This study presents a statistical model to predict avalanche and non-avalanche days using a combination of weather data, modelled snowpack properties, and modelled stability indices. The model is developed with 58 years of avalanche observations from a region in France. The study is designed to examine the added value of stability indices in statistical models for avalanche activity. While statistical models have been widely developed and tested in the scientific literature, investigating how recent advances in snowpack modelling and snow mechanics could improve these models is an interesting and worthwhile objective that is well suited for The Cryosphere. My main concern is how some of the methodological choices likely impacted the results and conclusions. I also think the study missed an opportunity to present their spatially distributed results (i.e., by aspect and elevation) which could be of value to avalanche forecasters. Please see my specific comments for suggested revisions to this paper.

We thank S. Horton for his detailed and constructive review. We answer point by point to all his comments hereafter.

**Specific comments**

- Manuscript structure: The paper was well structured with complete and logical flow of information. The graphics were also clean and easy to interpret.
- Sampling of days to include in the study: I question some of the choices made about filtering the data set and how that impacted the results. A few things stand out as dramatically impacting the set of avalanche days and non-avalanche days that were analyzed:
  - Why was the period restricted to Oct 15 to Mar 15? Doesn't this remove a large portion of large wet avalanches from the study? What is the purpose of including wet snow stability indices when many of the wet snow avalanche days have been removed? Do you have any information about wet versus dry avalanche activity in the EPA data set? Similarly, I question how meaningful including days in October and November are for predicting full path avalanches.

The period is from 15 Oct to 15 **May**. We will correct this error in our original version in the revised manuscript. The choice of dates inevitably contains arbitrariness. Our goal was to include a large variety of situations. In France, the avalanche bulletin is produced from early November to early June, which is coherent with the selected date range. Our choice is also coherent with the choice of other studies, such as [Sielenou, 2021], for instance. We will specify these reasons in the text.

-
  - Second, the threshold of 1 cm (or 10 cm in other parts of the manuscript?) seems very low considering the avalanche observation data only considered avalanches reaching the bottom of avalanche paths. I think a larger threshold would be much more appropriate. Choosing a threshold depth for avalanches grounded in literature or deriving one form your data set would be more appropriate (e.g., calculate the distribution of snow depths on avalanche days and chose a low percentile as a cut-off). I assume this would be on the order of 100 cm and would remove many of the non-avalanche days from the study.

Sorry for the inconsistency. We will correct the value line 186. We use a threshold of 10 cm to remove days with no or few snow on ground. The threshold is inevitably arbitrary. We would like to keep all situations that could lead to avalanches and therefore select a conservative value. It therefore allows for a significant undersampling, especially on low elevation bands. The statistical algorithm then have the role of selecting optimal values to separate between avalanche and non avalanche situations. Hence, this threshold is chosen to be conservative and not optimal in any

way. We will summarize this in the text.

- - I suspect plotting the avalanche activity by day of year and snow depth would reveal informative patterns about when discriminating avalanche and non-avalanche days is actually important to avalanche forecasters. A model informing the likelihood of large natural avalanches in mid-winter and late-winter is likely much more helpful than a model informing whether the snowpack depth has reached the threshold for avalanches.

We work on 58 years and 24 aspect-elevation bands. It would be impossible to visually interpret something from this large amount of data. Moreover the central question of this paper is on the interest of physically-based stability indicators to summarize information from snow cover models before applying machine learning methods. We thus propose, according to the proposition of an other reviewer, to provide an example of one representative output for one aspect and elevation. This will not provide a systematic evaluation of the interest for forecasters, that is done by the overall scores but allow for a qualitative interpretation. For instance, this will highlight that the model do not only inform on the fact that a sufficient snowpack depth is reached.

- - By removing more of the uninteresting non-avalanche days, the dataset would be more balanced. This would likely diminish the obvious impacts of snow depth on the resulting models and put more weight on the stability indices, which would better suit the objective of the study.

We do not currently have elements to justify that the weight of snow depth is linked to unbalancing. All previous studies show a high importance of snow depth, new snow depth or overall precipitation, despite they use different levels of unbalancing (e.g. Davis et al, 1999; Hendrikx et al., 2014; Scwheizer et al., 2009; Sielenou et al., 2021). The new snow depth is also the first criterion for most practitioners and forecasters for natural avalanche activity.

- Weak layer selection: The choice of always selecting 5 weak layers seems unusual and was not adequately justified. What is the benefit to this method over choosing a threshold value to identify weak layers? Could there be adverse effects to having many extra layers in the analysis that are potentially stable and uninteresting? For example, wouldn't this diminish the importance of the stability indices compared to a dataset that only included layers that met some type of threshold stability criteria?

The goal of the Random Forest technique is to optimize thresholds to decide whether a weak layer should be considered as prone to avalanche or not, by considering the values of its different stability indices as well as its depth in the snowpack. In this study, the goal is to extract the potentially relevant information from the snow cover model that may not be available to statistical tools otherwise (as nobody will try to put all the output of a snow cover model as an input of statistical model) and then let the statistical method define, from observations what is relevant or not. We therefore do not want to introduce expert thresholds on the different stability indices. Moreover, we need a constant input parameter number, even though we are not able to find a relevant weak layer in the snowpack. We thus selected 5 weak layers as we have five stability indices that may point out a weak layer. It may lead to redundancy of the information in some cases. We already have a lot of redundancy (correlation) in our input variables. We will add a sentence to justify this choice: *This approach allow for identifying the five weakest layers, with five complementary ways of estimating the weakness (five stability indices), and have the advantage to provide a constant number of data for further statistical analysis.*

- Classification scores and model performance: I wonder how my previous comments impact the resulting classification scores. The precision seems very low, despite the explanation provided. I was also surprised to see the low performance of the meteo subset, as I would expect weather factors to be significantly better at predicting natural avalanche activity than a random model. Especially when considering large natural avalanches, common forecasting experience and past studies have found simple weather indices like 72 hour accumulated precipitation and air temperature to be strong influences. This has me question the representativeness of the dataset/variables and the overall soundness of the results. Can you justify the low performance of the meteo subset in this model?

In this study we explore a large time period and detail the results by aspect and elevation bands. Meteorological variables are highly correlated between elevation and aspect sectors (except incoming radiations that are not in the "Meteo" variable set here). However, in early season or in spring, lower elevations are no longer prone to avalanche while higher elevation may experience higher activity. Weak layer may also form differently depending on aspect. Therefore, we expected meteorological variables to be insufficient to describe the avalanche activity in aspect-elevation sectors. We will introduce a sentence in the discussion to precise this results.

- No presentation of results by aspect and elevation: While I understand the decision to aggregate the results from different aspect and elevations to see the overall importance of input variables, I think presenting some of the aspect and elevation patterns would be of great interest as well. First, the question of how well the model can predict the location of avalanche activity would be valuable to forecasters. Second, it's not clear whether the imbalance in the amount of avalanche days by terrain class shown in Fig. 2 impacted the results (e.g., how does the model performance compare on south aspects where there were many avalanche days versus NE aspects where there were few avalanche days).

On the first question of how well the model predicts the location (in a semi-distributed model where location means altitude and elevation) of avalanches, we answer the question as we provide scores for the identification of avalanche activity by classes of elevation and aspect even though only the overall results are presented. We do not plan

On the imbalance of the avalanche activity we present, it depends on the goal of the study. As we point out in the description of the dataset, we believe that due to the high number of observed avalanche paths and the steepness of the slopes, the recorded EPA avalanche activity is a good proxy of overall avalanche activity of the Haute-Maurienne valley. Hence, when the goal is to predict the avalanche activity of Haute-Maurienne, the use of a realistic avalanche activity, including unbalance between elevation and aspects seem relevant. However, this means that the model may not be directly transferable to other areas. We will introduce this in the discussion: *we here train the model with the Haute-Maurienne data. Some climatological or terrain features may lead to a predicted avalanche activity specific to the Haute-Maurienne area, especially with a higher senitivity of certain aspects or elevations (eastern crests during easterly returns). The model may not be transferable directly to other areas without a new calibration.*

Moreover, raw results by band of altitude and orientation are difficult to interpret because each aspect and elevation have different number of avalanche observations involved. In a sector where only one avalanche was observed, the score can be either 0 or 1 but this does not carry any information. We nevertheless checked that there is no obvious over- or under-performance for sectors with significantly more or less observations.

- Writing style: I found parts of the manuscript difficult to read, with poor flow between sentences and phrases interrupted by citations. I had to read some paragraphs twice to fully understand the meaning and would appreciate additional editing to improve the readability.

We will carefully reread the manuscript to improve the overall flow.

**Technical comments**

We thank the reviewer for the detailed comments. We answer to comments that will not be directly taken into account as proposed or ask for more explanations.

- Title: Is "snow physics" the best way to describe the dataset in this study? It has a broad range of interpretations and when first reading the manuscript I wouldn't have automatically assumed the main data was model-generated stability indices.

With the suggestion of all reviewers, we will change the title to *Does combining modelled snowpack stability with machine learning help with predicting avalanche activity?*

- Lines 11-12: The terms "recall" and "precision" are rather technical for the abstract and would probably have more impact if replaced with plain language descriptions (e.g., predicted X% of days when avalanches were observed), especially considering there are many synonyms for contingency table statistics and some readers may not be familiar with these specific ones.

We will edit the abstract with your suggestion.

- Line 20 "Human infrastructure" is an unusual term and could probably be described better.
- Line 19-23: These first few sentences are examples where the position of citations interrupts the readability.
- Line 42: The phrase "delimitation lines around avalanche-prone conditions" is verbose and could be more concise and clear.
- Lines 50-52: Nice context and motivation for this study!
- Line 52: I question whether adding mechanical stability indices would "reduce the complexity of statistical

> tools". These tend to be relatively complex variables dependent upon many other parametrized variables, and in my view are more complex than a simple model based on variables like snow depth and air temperature. I suggest removing "reduced complexity" and directly stating what is meant by complexity (i.e., models with fewer variables and interactions).

We propose to rephrase this sentence and add "*reduced complexity compared to a model that would directly use the snow cover model output*".

> - Lines 62-63: This important sentence stating the objective of the study should be written to be more clear and specific. I had to read this multiple times and was still unclear on the big picture aim of the study.

We propose to reformulate this sentence as follows: *On this basis, the aim of this paper is to determine if the combination using machine learning techniques of observed avalanches data and mechanical stability analysis of snow profiles helps for predicting avalanche activity.*

> - Line 72: remove "an" from "study an area"
> - Line 75: What is meant by a "series of events" being reliable? Is this refereeing to reliable observations of the events?

Yes, we will update the text to make it clearer.

> - Line 80: Please justify this date range. As mentioned above, the early part of this range likely contains many uninteresting non-avalanche days and the late part of this range omits large spring avalanches. This date range criteria could be dramatically influencing the results and their interpretation.

The period is from 15 Oct to 15 **May**. We will correct this error in our original version in the revised manuscript. The choice of dates inevitably contains arbitrariness. We would like to include a large variety of situations. In France, the avalanche bulletin is produced from early November to early June, which is coherent with the selected date range. Our choice is also coherent with the choice of other studies, such as [Sielenou et al., 2021], for instance. We will specify these reasons in the text.

> - Line 88: Can you comment on the typical size of these avalanches that reach the run-out threshold (e.g., using the EAWS scale https://www.avalanches.org/standards/avalanche-size/). This would help readers better understand the type of avalanches this model predicts. Also, are all these avalanches natural or are any of the paths modified or controlled with explosives (because snowpack would impact the representativeness of the snowpack model)?

The observation network was designed at the end of the 18th century when avalanche sizes were not yet normalized. Hence, the avalanche size is not explicitly used. The minimal size is indirectly defined for each avalanche path by the position of the observation threshold that should be crossed by avalanches to be recorded by the observer. Empirically, we can imagine that no size 1 avalanche are recorded. Avalanches of size 2 may be recorded, especially if an accident is related to this avalanche or if the avalanche reached high altitude infrastructure and most of the recorded avalanches may be of size 3 or more. However, this was never evaluated. We thus prefer not to give an indication that will not be properly supported. Moreover, due to the high number of avalanche paths in Haute-Maurienne and the steepness of the slopes, we believe that EPA provides a good overview of avalanche activity, as we point out in the description of the dataset: *Besides, the steep topography of Haute-Maurienne reduces the effect of the threshold of observation as most of the avalanches reach the valley floor, providing a representative screenshot of avalanche activity of avalanches reaching low altitudes.*

> - Line 98: Please describe how avalanche date uncertainty is defined? Do observers estimate a range of dates?

Yes, uncertainty is estimated by observers based on their previous visit. We will add a precision on that in the text.

> - Line 116: Was the entire study area treated as a single massif in SAFRAN or was SAFRAN run for each municipality? If a single massif, why is it meaningful to show the three municipalities in Fig. 1?

We work on a SAFRAN massif from the simulation point of view. Observations used are those of the three selected districts. The three districts are plotted to delimit the overall studied area on the map and help the reader to locate the area of interest.

- Line 131: I think a bit more detail about these indices could be included in this section rather than referring to another paper. Providing equations and/or describing some of the key snowpack outputs used to calculate strength and stress would be valuable. Also, the only reference for Viallon-Galinier (2021) in the reference list is https://doi.org/10.1016/j.coldregions.2020.103163, but I think these citations are intended to refer to https://doi.org/10.1016/j.coldregions.2022.103596 which is not listed.

Sorry for the error on the citation, we will correct it in the revised version.

We will include a direct reference for each stability indices. We also explain the main principles of each stability indicator.

- Line 133: The choice to select five weak layers from every profile is not adequately justified. Also see my specific comment about how this may impact the results. Also, when defining the local minimum is one layer identified for each separate indices or is there some type of weighted average? If the former case, are there situations where a layer may be duplicated because it is the minimum for multiple indices?

See the main comment answer.

- Sect 2.4.3: I really like the addition of these time derivatives and think it is an interesting part of the study!
- Sect 2.5.1: With such a rich observation dataset I wonder why the simplest binary metric for avalanche activity was chosen. I would expect between the large set of avalanche observations and the types of stability indices included in the models you could try to predict more advanced indicators such as weighted avalanche activity indices, percentage of paths in an aspect-elevation sector that released, etc. The chosen indicator is fine, but perhaps the choice could be justified a bit more.

Most of the time, only few avalanches are observed in each aspect and elevation sector. Hence, the selected dataset and space partition do not allow to easily overcome the binary classification. However, we propose to add more discussion on this point as the combination with other data sources may allow to remove this limitation.

- Line 160: Be careful with using the term "the model" throughout the paper when both the physical snowpack model and statistical model are part of the study.

We will carefully check all occurrences to remove ambiguity when it exists.

- Table 1: I appreciate this concise summary of model inputs. Minor corrections are the depth of dry snow weak layers is listed in consecutive rows, units are provided in different columns, and column 2 is missing a title.
- Lines 180-190: Are there also concerns about the imbalance in the aspect-elevation data? For example, based on Fig. 1 and 2 I assume the number of start zones per sector are variable, so is it reasonable to have an equal number of data points for NE and S aspects in the analysis?

Avalanches are observed only if they are located in pre-defined avalanche paths and reach the observation line. However, information reported contain the elevation of departure of the avalanche and the aspect of the area where it started. This may not be directly linked to avalanche path information as a path may be globally south-oriented but sides may look south-east or south-west, or even East or West for avalanche paths with large departure zones. We use the information related to the precise recorded avalanche as soon as it is available (most of the time), that is why we presented the data for the observed avalanches rather than for the avalanche paths.

Moreover, as the valley turn, we have globally north and south facing paths in Lanslevillard and more South-East and West facing path in Bessans for instance. However, even though a wide variety of aspects is represented in terms of avalanche paths, this does not ensure that all paths are equivalent. Some may have more forest than other, or different vegetation influencing susceptibility to avalanches, some may be steeper than others, etc. We cannot ensure an equal representation of all possible aspects and elevations with similar conditions.

As we point out in the description of the dataset, we believe that due to the high number of observed avalanche paths and the steepness of the slopes, the recorded EPA avalanche activity is a good proxy of overall avalanche activity of the Haute-Maurienne valley. Hence, when the goal is to predict the avalanche activity of Haute-Maurienne, the use of a realistic avalanche activity, including unbalance between elevation and aspects seem relevant. However, this means that the model may not be directly transferable to other areas. We will introduce this in the discussion: *we here train the model with the Haute-Maurienne data. Some climatological or terrain features may bias the avalanche activity*

*towards certain aspects or elevations (eastern crests during easterly returns). The model may not be transferable directly to other areas without a new calibration.*

- Line 185: A 1 cm threshold seems very small for full path avalanches.

The threshold is 10 cm. We will correct it in the revised version. This threshold inevitably contains some arbitrariness. However, the optimization of the Random forest should be able to select the most relevant one, we just remove days for which we are completely sure there is no interest of studying avalanche activity to prevent overwhelming of the optimization algorithm with uninteresting situations.

- Sect. 2.6: I like the LOYO validation approach used in this study and it is well described here. One minor comment is why was the 20 to 80th percentiles chosen when 25-75, 10-90 or 5-95 percentile ranges are more common?

We have 58 years. Therefore, the choice of a classical 5-95 percentile would require additional assumptions on the statistical repartition of the errors for the computation. The 20-80 percentile does not seem completely unusual and suited to our dataset.

- Fig. 3: Please specify the range of uncertainty in the caption (i.e., 20th to 80th percentile).

We will specify the uncertainty range in the caption.

- Line 260: Here and in Fig. 4 a new way of grouping the variables is introduced which differs from Table 2. I can track how these counts arise, but it could be clearer.

We will clarify this.

- Line 257: What is meant by new snow variations? This sounds like change in snow depth, which is not a variable listed in Table 1. Also, I would consider separating the snow depth from variations in Fig. 4 to see how much of the predictive power was simply due to snow depth reaching the threshold for avalanches versus how much was due to detecting snow depth changes over shorter time intervals.

The sentence will be rewritten as *new snow amounts or snow depth variations*.

- Fig 4: I suggest sorting the rows by WSSI and DSSI rather than time step to more clearly show the impact of different step sizes.

We will adapt as suggested, as it will improve the readability of the figure.

- Line 294: Please the describe the context referred to in Rubin et al. (2012), I am curious how such low precision has been justified in other studies rather than highlighting some type of issue with how the study was designed.

The value of the precision is highly linked to the unbalancing of the dataset. We have more than 233 000 non-avalanche situations while only 2608 avalanche situations. If we artificially reduce the number of non-avalanche situations in the test dataset (e.g. by randomly drawing observations), we will increase the precision value while not changing anything to the capabilities of the model. We believe that we provide values that are close to real use case of such model as avalanche situations are far from majority of situations.

- Line 300: I disagree that the obvious non-avalanche days have been removed (see Specific comments).

We will rephrase to say that we removed some of the most obvious non-avalanche days.

- Lines 310-318: While I understand how the model is build with aspect-elevation specific inputs, I think presenting some of the terrain specific results would be a highly interesting part of the study.

We plan to publish in-depth evaluation of the interest of this AE approach in a different paper. We focus in this paper on the question of the interest of stability indices (and derivatives) in combination with machine learning. We thus think that specific situations will not improve the readability and clarity of this paper.

---

## Referee Report (RR1)

**General comments**

I appreciate the thoughtful revisions to this manuscript which have clarified many of my initial concerns. Many of the methodological choices are now clearer and better put the results into context. In particular, the explanation of how of a correctly predicting avalanche activity in specific elevation and aspect terrain classes is more difficult that predicting activity at a regional scale, is much clearer. With these rationales clearer I still have some concerns about how the variable groupings impact the results and a few suggestions to strengthen some arguments. Thanks, it was an enjoyable and interesting manuscript to read.

**Specific comments**

- **Variable groupings.** I find the limited choice of meteorological variables misleading, especially when this group performs similar to a random classifier. First, not all the meteorological inputs appear to be included, especially ones that would specifically improve the prediction at the spatial resolution investigated. For example, solar radiation and wind direction should be primary drivers of snowpack variability across different aspects and precipitation, temperature and wind speed should be driving variability across elevations. The resulting snowpack properties on different terrain classes are ultimately a product of the meteorological inputs, and by not including these in the analysis it is not a representative comparison of a model with and without stability indices. With these groupings some of the predictive skill being attributed to stability indices does not truly reflect the added value of a snowpack model. Similarly, including snow depth change in the stability group may also inflate the importance of stability indices. First, based on many previous studies this is expected to be one of the most important predictors of avalanche activity. Second, it would make more sense to consider this a meteorological variable or, specifically in this study, a derivative of a bulk variable. While I understand the argument that the current structure shows how stability indices aggregate the information in a way that explains the dataset well, the goal of the study suggested by the title is to ask "does it help?", which I think requires a precisely thought out structure that compares the ability to predict avalanche-situations with and without stability information.
- **Variable importance.** The way the results of the Gini importance are presented in Fig. 5 make it difficult to follow the discussion in lines 277-286, because individual variables are discussed in the text but only the group values are shown in the figure. For example, although snow depth is reported to be the most important, its value is not reported in the text and the group of dry snow stability indices collectively have more importance in the figure. I think these results could be presented in a clearer and more consistent way. In some ways this relates to the previous comment about some arbitrary groupings. Perhaps presenting the Gini importance values for every variable in an appendix would help, perhaps sorted by importance within each group.
- **Oversample low snow depth days.** Initially from a forecasting perspective the 10 cm threshold to remove non-avalanche situations seems overly conservative. My concern is the importance of snow height would dominate over any influence of stability indices, especially in a dataset dominated by full path avalanches. After reviewing the results in more detail, I am now satisfied with this concern by seeing that the model with stability indices alone, without snow depth, perform at a similar level and thus must capture the important influence of snow depth some

way. I'm wondering if this specific point should be emphasized more to highlight that the impact of threshold snow depth can be captured within the set of stability variables.

- **Impact of aspect-elevation resolution on performance.** Can you explicitly state why the aspect-elevation resolution leads to lower performance due to more precise resolution? I assume it's because it is harder to predict the correct aspect and elevation of an avalanche than predict an avalanche anywhere in a region. Providing a direct explanation of this in the discussion would strengthen the argument that the some of the low performance metrics are justified.

**Technical comments**

- Line 97-98: Can you clarify what is meant by "observation threshold"? Perhaps this could be clarified by being more specific in the line 94 with "run-out reached a certain threshold distance" and then in line 97-98 sentence make it clear "although the observation data only includes avalanches that reach the threshold distance, the dataset provides an overall representative indication of avalanche activity in the area because the steep topography of the Haute-Maurienne causes most avalanches to reach this distance…"
- Table 1: Why is snow depth change in stability?
- Line 196: Is there a word missing in "goal is not to substitute to the machine learning algorithm"? The sentence is unclear.
- Line 278-279: Why are the Gini importance values not reported for these most importance variables, but are reported for subsequent groups? The reporting of these results is inconsistent and difficult to interpret from Fig. 5.
- Sect. 4.2: It would be interesting to report which specific stability indices had the highest relative importance. Why are these not presented? This could be interesting for future reach into stability indices.

**Initial review**

**Editor**

Dear Jurg Scweizer,

I found the study interesting however have several concerns about some of the methods and results that I recommended be addressed in a revision.

Best, Simon Horton

**General comments**

This study presents a statistical model to predict avalanche and non-avalanche days using a combination of weather data, modelled snowpack properties, and modelled stability indices. The model is developed with 58 years of avalanche observations from a region in France. The study is designed to examine the added value of stability indices in statistical models for avalanche activity. While statistical models have been widely developed and tested in the scientific literature, investigating how recent advances in snowpack modelling and snow mechanics could improve these models is an interesting and worthwhile objective that is well suited for The Cryosphere. My main concern is how some of the methodological choices likely impacted the results and conclusions. I also think the study missed an opportunity to present their spatially distributed results (i.e., by aspect and elevation) which could be of value to avalanche forecasters. Please see my specific comments for suggested revisions to this paper.

**Specific comments**

- **Manuscript structure:** The paper was well structured with complete and logical flow of information. The graphics were also clean and easy to interpret.
- **Sampling of days to include in the study:** I question some of the choices made about filtering the data set and how that impacted the results. A few things stand out as dramatically impacting the set of avalanche days and non-avalanche days that were analyzed:
    - Why was the period restricted to Oct 15 to Mar 15? Doesn't this remove a large portion of large wet avalanches from the study? What is the purpose of including wet snow stability indices when many of the wet snow avalanche days have been removed? Do you have any information about wet versus dry avalanche activity in the EPA data set? Similarly, I question how meaningful including days in October and November are for predicting full path avalanches.
    - Second, the threshold of 1 cm (or 10 cm in other parts of the manuscript?) seems very low considering the avalanche observation data only considered avalanches reaching the bottom of avalanche paths. I think a larger threshold would be much more appropriate. Choosing a threshold depth for avalanches grounded in literature or deriving one form your data set would be more appropriate (e.g., calculate the distribution of snow depths on avalanche days and chose a low percentile as a cut-off). I

assume this would be on the order of 100 cm and would remove many of the non-avalanche days from the study.

  o I suspect plotting the avalanche activity by day of year and snow depth would reveal informative patterns about when discriminating avalanche and non-avalanche days is actually important to avalanche forecasters. A model informing the likelihood of large natural avalanches in mid-winter and late-winter is likely much more helpful than a model informing whether the snowpack depth has reached the threshold for avalanches.

  o By removing more of the uninteresting non-avalanche days, the dataset would be more balanced. This would likely diminish the obvious impacts of snow depth on the resulting models and put more weight on the stability indices, which would better suit the objective of the study.

- **Weak layer selection:** The choice of always selecting 5 weak layers seems unusual and was not adequately justified. What is the benefit to this method over choosing a threshold value to identify weak layers? Could there be adverse effects to having many extra layers in the analysis that are potentially stable and uninteresting? For example, wouldn't this diminish the importance of the stability indices compared to a dataset that only included layers that met some type of threshold stability criteria?

- **Classification scores and model performance:** I wonder how my previous comments impact the resulting classification scores. The precision seems very low, despite the explanation provided. I was also surprised to see the low performance of the meteo subset, as I would expect weather factors to be significantly better at predicting natural avalanche activity than a random model. Especially when considering large natural avalanches, common forecasting experience and past studies have found simple weather indices like 72 hour accumulated precipitation and air temperature to be strong influences. This has me question the representativeness of the dataset/variables and the overall soundness of the results. Can you justify the low performance of the meteo subset in this model?

- **No presentation of results by aspect and elevation:** While I understand the decision to aggregate the results from different aspect and elevations to see the overall importance of input variables, I think presenting some of the aspect and elevation patterns would be of great interest as well. First, the question of how well the model can predict the location of avalanche activity would be valuable to forecasters. Second, it's not clear whether the imbalance in the amount of avalanche days by terrain class shown in Fig. 2 impacted the results (e.g., how does the model performance compare on south aspects where there were many avalanche days versus NE aspects where there were few avalanche days).

- **Writing style:** I found parts of the manuscript difficult to read, with poor flow between sentences and phrases interrupted by citations. I had to read some paragraphs twice to fully understand the meaning and would appreciate additional editing to improve the readability.

**Technical comments**

- Title: Is "snow physics" the best way to describe the dataset in this study? It has a broad range of interpretations and when first reading the manuscript I wouldn't have automatically assumed the main data was model-generated stability indices.

- Lines 11-12: The terms "recall" and "precision" are rather technical for the abstract and would probably have more impact if replaced with plain language descriptions (e.g., predicted X% of days when avalanches were observed), especially considering there are many synonyms for contingency table statistics and some readers may not be familiar with these specific ones.
- Line 20 "Human infrastructure" is an unusual term and could probably be described better.
- Line 19-23: These first few sentences are examples where the position of citations interrupts the readability.
- Line 42: The phrase "delimitation lines around avalanche-prone conditions" is verbose and could be more concise and clear.
- Lines 50-52: Nice context and motivation for this study!
- Line 52: I question whether adding mechanical stability indices would "reduce the complexity of statistical tools". These tend to be relatively complex variables dependent upon many other parametrized variables, and in my view are more complex than a simple model based on variables like snow depth and air temperature. I suggest removing "reduced complexity" and directly stating what is meant by complexity (i.e., models with fewer variables and interactions).
- Lines 62-63: This important sentence stating the objective of the study should be written to be more clear and specific. I had to read this multiple times and was still unclear on the big picture aim of the study.
- Line 72: remove "an" from "study an area"
- Line 75: What is meant by a "series of events" being reliable? Is this refereeing to reliable observations of the events?
- Line 80: Please justify this date range. As mentioned above, the early part of this range likely contains many uninteresting non-avalanche days and the late part of this range omits large spring avalanches. This date range criteria could be dramatically influencing the results and their interpretation.
- Line 88: Can you comment on the typical size of these avalanches that reach the run-out threshold (e.g., using the EAWS scale https://www.avalanches.org/standards/avalanche-size/). This would help readers better understand the type of avalanches this model predicts. Also, are all these avalanches natural or are any of the paths modified or controlled with explosives (because snowpack would impact the representativeness of the snowpack model)?
- Line 98: Please describe how avalanche date uncertainty is defined? Do observers estimate a range of dates?
- Line 116: Was the entire study area treated as a single massif in SAFRAN or was SAFRAN run for each municipality? If a single massif, why is it meaningful to show the three municipalities in Fig. 1?
- Line 131: I think a bit more detail about these indices could be included in this section rather than referring to another paper. Providing equations and/or describing some of the key snowpack outputs used to calculate strength and stress would be valuable. Also, the only reference for Viallon-Galinier (2021) in the reference list is https://doi.org/10.1016/j.coldregions.2020.103163, but I think these citations are intended to refer to https://doi.org/10.1016/j.coldregions.2022.103596 which is not listed.
- Line 133: The choice to select five weak layers from every profile is not adequately justified. Also see my specific comment about how this may impact the results. Also, when defining the local minimum is one layer identified for each separate indices or is there some type of weighted average? If the former case, are there situations where a layer may be duplicated because it is the minimum for multiple indices?

- Sect 2.4.3: I really like the addition of these time derivatives and think it is an interesting part of the study!
- Sect 2.5.1: With such a rich observation dataset I wonder why the simplest binary metric for avalanche activity was chosen. I would expect between the large set of avalanche observations and the types of stability indices included in the models you could try to predict more advanced indicators such as weighted avalanche activity indices, percentage of paths in an aspect-elevation sector that released, etc. The chosen indicator is fine, but perhaps the choice could be justified a bit more.
- Line 160: Be careful with using the term "the model" throughout the paper when both the physical snowpack model and statistical model are part of the study.
- Table 1: I appreciate this concise summary of model inputs. Minor corrections are the depth of dry snow weak layers is listed in consecutive rows, units are provided in different columns, and column 2 is missing a title.
- Lines 180-190: Are there also concerns about the imbalance in the aspect-elevation data? For example, based on Fig. 1 and 2 I assume the number of start zones per sector are variable, so is it reasonable to have an equal number of data points for NE and S aspects in the analysis?
- Line 185: A 1 cm threshold seems very small for full path avalanches.
- Sect. 2.6: I like the LOYO validation approach used in this study and it is well described here. One minor comment is why was the 20 to 80th percentiles chosen when 25-75, 10-90 or 5-95 percentile ranges are more common?
- Fig. 3: Please specify the range of uncertainty in the caption (i.e., 20th to 80th percentile).
- Line 260: Here and in Fig. 4 a new way of grouping the variables is introduced which differs from Table 2. I can track how these counts arise, but it could be clearer.
- Line 257: What is meant by new snow variations? This sounds like change in snow depth, which is not a variable listed in Table 1. Also, I would consider separating the snow depth from variations in Fig. 4 to see how much of the predictive power was simply due to snow depth reaching the threshold for avalanches versus how much was due to detecting snow depth changes over shorter time intervals.
- Fig 4: I suggest sorting the rows by WSSI and DSSI rather than time step to more clearly show the impact of different step sizes.
- Line 294: Please the describe the context referred to in Rubin et al. (2012), I am curious how such low precision has been justified in other studies rather than highlighting some type of issue with how the study was designed.
- Line 300: I disagree that the obvious non-avalanche days have been removed (see Specific comments).
- Lines 310-318: While I understand how the model is build with aspect-elevation specific inputs, I think presenting some of the terrain specific results would be a highly interesting part of the study.

---

## Editor Decision (ED1)

**Editor's comments**

«Does combining modelled snowpack stability with machine learning help with predicting avalanche activity?"

submitted by Viallon-Galinier, Hagenmuller and Eckert to *The Cryosphere*

The paper describes a study combining weather data, modelled snow data and modeled snowpack stability data with observations of avalanches employing the random forest method to forecast avalanche activity (avalanche/non-avalanche days) for 24 for elevation and aspect segments.

The manuscript is easy to follow; the research objectives are relevant and doubtless challenging. The manuscript is a valuable contribution to the field of numerical avalanche forecasting.

As reviewers have pointed out – and I share their concerns – it is questionable whether the avalanche dataset selected is fully suitable to reach the goals of the study. With less than 3000 avalanches in 58 years observed in 110 avalanche paths, about every second winter an avalanche is observed in a specific path. Whether this frequency is suitable to predict daily avalanche activity in 24 aspect and elevation segments remains to be shown.

This said I do neither oppose the use of the dataset and nor question the study overall, I simply invite the authors to reflect whether the question put in the title can be adequately answered and how well the results can be generalized.

On the title, for instance, my recommendation is not to word it as a question, or at least put the question in a way that it can be answered. For instance, to answer the question you would need to consider combinations other than stated in the title. However, you only explore one method. I suspect you primarily wonder about the value of modeled stability information (compared to, for instance, weather and snow data only), in particular in view of your goal to forecast for 24 aspect and elevation segments. By the way, stability information, mostly not modelled, has been used in several previous studies, even back in some early work on numerical forecasting in the 1990ies – and was shown to be important.

Below you will find some more specific comments:

| | |
|---|---|
| Lines 13-14: | I think it would be valuable to show how the model would perform with a simple target variable (not considering 24 aspect and elevation segments). Moreover, I think "cutting-edge" should not be related to data here. |
| Lines 20-21: | I recommend rewording. Long-term and short-term are somewhat oddly used here. I suppose you refer to hazard assessment in the context of hazard mapping vs. avalanche forecasting. Hazard mapping and avalanche forecasting are both mitigation measures having a long-term and a short-term effect, respectively. |
| Line 24: | I am not sure whether you focus on forecasting or prediction. |
| Line 34: | Please cite the corresponding peer-reviewed publication rather than a magazine article (Schweizer and Jamieson, 2007). |
| Line 53: | Some early models that used stability information were described by Schweizer et al. (1994) and Schweizer and Föhn (1996). |
| Line 58: | Schirmer et al. (2009) used output variables from a numerical snow cover model as input for statistical forecasting. |

Line 65: It has been shown that the type of data (input variables) can at least be as important as the method applied. For instance, regardless of the method, with meteorological data as input only, the performance is always limited (e.g., Schirmer et al., 2009)

Lines 66-67: Suggest rewording to better reflect the aim.

Lines 67-68: Suggest rewording so that the three different types of input data become more obvious.

Line 79: In your replies to the reviewers' comments, you point out many uncertainties related to the dataset. Hence, I wonder whether you can state that the avalanche observations are particularly reliable.

Lines 97-98: As you describe above, EPA includes primarily large natural avalanches in selected path. It is questionable whether these data describe the overall avalanche activity in the area. There is much more potential avalanche terrain in the area and almost always when there are large or very large avalanches, there will be many small and medium-sized avalanches as well. On the other hand, I suspect, there are also numerous days when no very large natural avalanches occurred, but still many small to medium-sized natural avalanches at higher elevations. Hence, what is observed and recorded in EPA cannot provide a complete picture of natural avalanche activity. All these additional smaller natural avalanches are very likely also relevant for operational avalanche forecasting. Hence, in conclusion, I suggest you reword and specify the statement.

Line 133: I recommend you add "for a given depth" or "a given layer interface"

Line 146&147: I recommend replacing humid by wet or moist.

Line 156: As far as I remember, Reuter et al. (2022) have recently applied the a time-dependent index to describe natural failure initiation.

Line 160: It is confusing to the reader that changes in snow depth are associated with stability indices. Hence, I suggest regrouping the input variables.

Line 166: I agree that the model resolution you select is more demanding. However, the question remains unanswered whether this additional sophistication represents an added value given the overall rather poor performance of the model from an operational point of view.

Lines 174-175: I suggest using more descriptive variable category names, for instance, weather, snow and stability. Certainly, "Bulk" does not seem appropriate to me. Also, in Table 1, you refer to derivates, which seem to include snow depth changes (though you only refer to dry snow indices). Please clarify.

Lines 219-221: I suggest you also mention that the recall is, commonly in the forecasting context, called the probability of detection (POD) or hit rate. This would ease comparison with previous work.

Lines 221-222: I suggest you also mention that the false positive rate is called the false alarm rate (FAR).

Line 236: Random classification is usually associated with the diagonal in the ROC diagram.

Line 254: elevation

Lines 257-258: The example you provide is probably one of the most prominent avalanche winters in your dataset. There were three, well-known major avalanche cycles in late January and

February in 1999. It would certainly be interesting to know the performance in such a "catastrophic" winter. In addition, are these model results obtained with the model trained with all input variables?

Lines 262-263: Please adapt the caption in Figure 4a to the description provided here.

Line 270: It may be worth mentioning that sensitivity (75.3%) and specificity (100-23.6%) are similarly good, which is nice, but that due to the unbalanced dataset the precision is low (3.**3**%).

Lines 278-279: The text here, in the revised manuscript, is unchanged, but Figure 5 (previously Figure 4) changed, i.e. the importance scores are different now. Are all changes in Figure 5 reflected in the text?

Line 280: In Figure 5 it is indicated that there are 25 variables from the dry snow stability group whereas in the text you refer to 30.

Line 294: Here you denote snow depth and its derivatives as "bulk", in contrast to the variable categories described in Table 1. Please clarify and/or improve Table 1.

Lines 313-314: What means "e.g. 2021"? By the way, both studies are published now. Please update the references (Mayer et al., 2022; Pérez-Guillén et al., 2022).

Line 336. I am not aware of any country who explicitly specifies the avalanche danger in 24 aspect and elevation segments.

Lines 347ff: It seems odd to me not to consider snow depth and in particular new snow depth as meteorological variables. In any case, you cannot conclude that meteorology is irrelevant and that this contrasts with other studies that probably all considered new snow depth. Moreover, it is doubtful that the alleged difference stems from modelled vs. measured new snow depth. Most readers would probably agree that new snow depth (precipitation) is a meteorological driver of avalanche danger. Hence, it seems that your conclusion depends on your choice of categories and cannot be generalized.

Line 353: Isn't new snow depth the significant variable rather than snow depth?

Lines 559-360: I am not sure how this last sentence refers to the importance of new snow depth.

Lines 371-373: As far as I remember van Herwijnen et al. (2016) and van Herwijnen et al. (2018) showed that dry-snow avalanche activity was correlated with snowfall on times scales of 2 or more days, while for wet-snow avalanches the correlation was shorter and with energy input. Hence this seems to be in contrast with what you describe. However, I agree with your statement in the following sentence.

Line 383: Previously you referred to 2779 events, not 2518. Please clarify.

Lines 384-285: Please refer to my previous comment on "the representative screenshot of the overall avalanche activity". In addition, I am not sure I understand why in Haute Maurienne the scarcity of reported avalanche events is not a problem.

Line 412: I am not sure I understand what you mean here with "the interest of physics".

Lines 419-420: Please reword.

Line 429: In my understanding "avalanche prediction" means to predict the exact time and location of a single avalanche event. I think so far, we rather forecast avalanche activity.

Lines 434-435:  I suggest rewording the statement: the combination proves to be valid.

Line 435:  I guess new snow depth was a significant variable, not snow depth.

Line 436:  As far as I remember, Jamieson and co-workers have shown that stability indices, though derived from measurements (not modeled), were valuable inputs for forecasting (e.g., Zeidler and Jamieson, 2004).

Line 439-440:  While I agree that snow cover models and thereof derived stability information is valuable, I recall that your model is not that great in identifying avalanche prone situations. The precision is low. Please consider rewording.

Line 443:  I suggest deleting "cutting-edge". The random forests method has become a rather standard tool over the course of the last decade. In addition, I am not sure you can call the data cutting edge.

Line 446:  Similarly, not convinced the stability indices are that cutting-edge.

Davos, 5 January 2023
Jürg Schweizer.

*References*

Mayer, S., van Herwijnen, A., Techel, F. and Schweizer, J., 2022. A random forest model to assess snow instability from simulated snow stratigraphy. The Cryosphere 16(11): 4593-4615.

Pérez-Guillén, C., Techel, F., Hendrick, M., Volpi, M., van Herwijnen, A., Olevski, T., Obozinski, G., Pérez-Cruz, F. and Schweizer, J., 2022. Data-driven automated predictions of the avalanche danger level for dry-snow conditions in Switzerland. Nat. Hazards Earth Syst. Sci., 22(6): 2031-2056.

Reuter, B., Viallon-Galinier, L., Horton, S., van Herwijnen, A., Mayer, S., Hagenmuller, P. and Morin, S., 2022. Characterizing snow instability with avalanche problem types derived from snow cover simulations. Cold Reg. Sci. Technol., 194: 103462.

Schirmer, M., Lehning, M. and Schweizer, J., 2009. Statistical forecasting of regional avalanche danger using simulated snow cover data. J. Glaciol., 55(193): 761-768.

Schweizer, J. and Föhn, P.M.B., 1996. Avalanche forecasting - an expert system approach. J. Glaciol., 42(141): 318-332.

Schweizer, J. and Jamieson, J.B., 2007. A threshold sum approach to stability evaluation of manual snow profiles. Cold Reg. Sci. Technol., 47(1-2): 50-59.

Schweizer, M., Föhn, P.M.B., Schweizer, J. and Ultsch, A., 1994. A hybrid expert system for avalanche forecasting. In: W. Schertler, B. Schmid, A.M. Tjoa and H. Werthner (Editors), Information and Communications Technologies in Tourism, Innsbruck, Austria, 12-14 January 1994. Springer Verlag Wien, New York, pp. 148-153.

van Herwijnen, A., Heck, M., Richter, B., Sovilla, B. and Techel, F., 2018. When do avalanches release: investigating time scales in avalanche formation. In: J.-T. Fischer et al. (Editors), Proceedings ISSW 2018. International Snow Science Workshop, Innsbruck, Austria, 7-12 October 2018, pp. 1030-1034.

van Herwijnen, A., Heck, M. and Schweizer, J., 2016. Forecasting snow avalanches by using avalanche activity data obtained through seismic monitoring. Cold Reg. Sci. Technol., 132: 68-80.

Zeidler, A. and Jamieson, J.B., 2004. A nearest-neighbour model for forecasting skier-triggered dry-slab avalanches on persistent weak layers in the Columbia Mountains, Canada. Ann. Glaciol., 38: 166-172.

---

## Author Response (AR2)

**Answer to F. Techel comments**

Léo Viallon-Galinier        Pascal Hagenmuller        Nicolas Eckert

> Dear authors
>
> Thank you for responding and addressing the points raised by the reviewers. From my perspective, the manuscript addresses most of the points appropriately.

We thank Frank Techel for his detailed and constructive comments that allow improving the paper. We answer point by point below. The original review is reported in green and associated with the answers in black. Quotations from the paper are in italic font and proposed changes in purple italic font.

**Main comments**

> I have two points, which I still feel are insufficiently addressed in the Data, Methods, and/or Discussion sections:
>
> 1. The first point refers to the way the data is filtered/labeled, and here, particularly the uncertainty of the release date. The labeling as a function of the release date uncertainty is (l104-106):
>    - 0 days: the release date is labeled avalanche situation
>    - [1, 2, 3] days (24% of the data): the last possible date within this range is labeled as an avalanche situation
>      - 3 days (28%): these cases are removed After removing the cases with uncertainty >3 days, the data set used for analyses, therefore, contains about one-third of the 2500 avalanche situations, where release date uncertainty is [1, 2, 3] days.
>    - Is this +/-[1,2,3] days? - Please include this information.

The number of days refers here to the length of the period on which the avalanche can have occurred. For instance, if an observer reports that an avalanche has occurred between the 21st and 23rd of January in a given path, we consider that the uncertainty of the report is 3 days (<= 3 days) and we arbitrarily consider that the avalanche occurred on the 23rd of January. If the observer reports an avalanche between the 10th and 14th of February, the uncertainty is 4 days (> 3 days) and we do not include the avalanche event in our database. We now detail in the text the definition of the uncertainty as the *length of the period on which the avalanche can have occurred* and provide an example. The paper now reads:

*To associate meteorological and snow conditions to each observed avalanche, we remove observations with an uncertainty (length of the period on which the avalanche can have occurred) of more than three days on the release date, from the dataset. When the uncertainty is larger than one day, the last day of the period was defined as the day of the avalanche event. For instance, if an observer reports that an avalanche has occurred between the 21st and 23rd of January in a given path, we consider that the uncertainty of the report is 3 days (<= 3 days) and we arbitrarily consider that the avalanche occurred on the 23rd of January.*

> 1. - You describe that the last day of the respective period is labeled an avalanche situation (l104-105). Please provide the respective information on how the other days within this range are labeled after this statement (I guess they are labeled non-avalanche situations as indicated on l166-168). Add a sentence as to why this approach is warranted as the other days within the range of the release date uncertainty also have a reasonably high chance that the avalanche occurred then.

As you noted, the definition of avalanche situations is provided lines 166-168. The other days within the range are labelled depending on the presence of other observations. If no other observations are reported for this day (after filtering and attribution of dates), the day is labelled as a non-avalanche day.

- Please discuss that one-third of the release date labels were uncertain and that this will likely impact the classifiers' observed performance. In that regard, could you provide an indication of how many of the 623 missed avalanche situations (Table 4) were missed simply because of this release date uncertainty? Being more clear about these errors would help to interpret whether the model really has a low precision or whether the target variable is just rather noisy (errors in the labels). For instance, Brenner and Gefeller (1997) demonstrate nicely the impact of erroneous labels on observed precision values (referred to as positive predictive value in their paper). This doesn't only apply to the uncertainty in the release date but also on errors in the labels in general.

We agree that the uncertainty related to the avalanche release date may affect the scores presented in this paper. Advanced and dedicated statistical tools that could deal with this uncertainty should be employed but are out of the scope of this paper. We chose to consider events with moderate uncertainty ($<= 3$ days) as certain with an occurrence date arbitrarily chosen. We may have kept only certain events and non-events (days not covered by any period of avalanche occurrence). However, in this case, there are only very few non-avalanche situations, as some uncertainties can be up to several months.

For the precision value, the *precision is highly influenced by the base rate (proportion of avalanchee situations). Here, avalanche and non-avalanche situations are highly unbalanced* as explained in section 4.1. Considering a balanced evaluation set, we would have a precision of around 76%. We added this in the text to better illustrate the dependency of the precision to class balancing. The uncertainty on the date is a drawback of the dataset that is discussed, but is not the main reason of that explains the scores. We also checked that some variations around the definition of avalanche situations does not change significantly the results.

2. You include Aspect-Elevation-(AE)-segments (Figure 2) that hardly ever have any avalanche observations. For instance, A=NE, E=1800 m, has one avalanche recorded in 58 years x 210 days. The base rate of avalanche situations is <0.0001. In the same aspect, E=3000 m had no avalanche recorded, and E=2400 m had six avalanches. It is, therefore (basically) impossible to make correct positive predictions. - Why are there so few avalanches in this aspect? Is the number of start zones per AE segment much lower than for other AE segments (please provide this information somewhere, maybe in a plot similar to Figure 2)? Or are these avalanche paths particularly difficult to observe, and, hence, observations less reliable? Or are there other factors, which cause these differences? –> Please discuss the implications of including these aspects and how this will impact the observed classifier performance.

In this work, the AE segments are not associated with avalanche paths but with the starting zone of each avalanche event. In particular, avalanche paths could cover large start zones, with different elevations and aspects. Hence, it is not possible to provide a repartition of starting zones. We nevertheless keep this idea for further work, in which we can group avalanche paths to define areas of interest and then ensure a number of observed avalanche paths per modelled segment. Moreover, the Haute-Maurienne topography is specific as the main avalanche activity comes from easterly returns. Hence, the eastern avalanche paths are expected to have a higher avalanche activity, especially for dry snow avalanches. The two sides of the valley also have different aspects, uses, which finally leads to differences in the avalanche activity. This leads to different avalanche activity in the different AE sectors. We do not believe this is a problem by itself. We try here to predict the avalanche activity of Haute-Maurienne. Training on this specific dataset allows us to capture all these local effects that affect the Haute-Maurienne avalanche activity. In a second time, we will work on the extension of this method. If we want to train the method in one area and apply to another, we have to get rid of these local specificities but this is not the goal of this study. We discuss it in section 4.3: *we here trained the model with the Haute-Maurienne data. Some climatological or terrain features may lead to a predicted avalanche activity specific to the Haute-Maurienne area, especially with a higher sensitivity of certain aspects or elevations (e.g., during easterly returns). Hence, the model may not be transferable directly to other areas without a new calibration.*

Is is very tricky to compare the performances between aspects and orientation. As you pointed out, the base rate is different. However, we checked that the scores were not directly correlated to the number of observed avalanches. For instance, at 2400 m, the true positive rate is 80.7% and the false positive rate is 31.5% for aspect SE (207 observed avalanches) and respectively 79.4% and 30.7% for NW aspect (504 observed avalanches). At 3000 m, the true positive rate is 75.5% and the false positive rate is 34.4% whereas there are only 45 observed avalanche situations. The main goal of the paper is to evaluate the respective role of different input variables on the prediction performance. However, we ensure that that the model is able to perform an analysis based on the data provided and the result is not only linked to unbalance (at least when dealing with true positive rate and false positive rate, the

precision being, by construction, highly related to the balance between avalanche and non-avalanche situations).

> Given these two points (1, 2), I don't agree with the statement on l.444 "more representative scores compared to other studies". While I fully agree that you used a "robust and conservative" (l320) LOYO method to analyze the model performance, potential sources of error in the avalanche situation labels leave quite a few questions open. I am aware that it is in the nature of the observational data used that these inevitably have a fair amount of uncertainty in the labels (i.e. wrong release date, uncertainty with regard to observations of 'none'). While you make the reader aware of this (Sect. 2.2), I suggest taking up this point when discussing the results, particularly when discussing precision values as this is most impacted by the low base rate and such errors.

We removed *more representative* and replaced it with *robust and conservative* as proposed.

**Further remarks**

> Figure 3: please describe in the caption (or change the x-axis labels) whether the year-month labels indicate the beginning or the end of the month. Please scale the x-axes for the same interval [0,0.2] as in (b). I am not sure about the journal's requirements, but you may have to move the y-axis labels to the left. Consider adding a dashed horizontal line indicating the base rate of avalanche situations in the data set (about 1%). This would allow to easily see that the predicted probability, despite being rather low in absolute terms, is actually much higher than the base rate in some cases.

We adapted the legend to precise the x axis and reminded the base rate, which is also defined in the methods section. We adapted the figure to have a similar y-axis.

> l315: Consider rephrasing this sentence as precision strongly depends on the base rate (of avalanche days; see also Brenner and Gefeller, 1997).

We split the sentence to improve the clarity of the message.

> l336: replace "in many countries" with "in France". Not every warning service uses the three elevation bands 1800 m, 2400 m, and 3000 m.

We reworded the sentence to avoid this confusion. "This spatial resolution enables to capture the spatial distribution of the expected avalanche activity in one region. This latter information is crucial to evaluate and describe the avalanche danger at regional scale (Morin et al., 2019)."

> l375: add "or rain" after "with solar radiation" (rain is a strong driver of snowpack wetting and avalanche release, i.e. Conway and Raymond, 1993)

Added.

> I hope these comments are helpful.

> Kind regards,

> Frank Techel

**References**

> Brenner and Gefeller, 1997

> Conway and Raymond, 1993

**Answer to Editor comments**

Léo Viallon-Galinier        Pascal Hagenmuller        Nicolas Eckert

**Main comments**

> The paper describes a study combining weather data, modelled snow data and modeled snowpack stability data with observations of avalanches employing the random forest method to forecast avalanche activity (avalanche/non-avalanche days) for 24 for elevation and aspect segments.
>
> The manuscript is easy to follow; the research objectives are relevant and doubtless challenging. The manuscript is a valuable contribution to the field of numerical avalanche forecasting.
>
> As reviewers have pointed out – and I share their concerns – it is questionable whether the avalanche dataset selected is fully suitable to reach the goals of the study. With less than 3000 avalanches in 58 years observed in 110 avalanche paths, about every second winter an avalanche is observed in a specific path. Whether this frequency is suitable to predict daily avalanche activity in 24 aspect and elevation segments remains to be shown.
>
> This said I do neither oppose the use of the dataset and nor question the study overall, I simply invite the authors to reflect whether the question put in the title can be adequately answered and how well the results can be generalized.
>
> On the title, for instance, my recommendation is not to word it as a question, or at least put the question in a way that it can be answered. For instance, to answer the question you would need to consider combinations other than stated in the title. However, you only explore one method. I suspect you primarily wonder about the value of modeled stability information (compared to, for instance, weather and snow data only), in particular in view of your goal to forecast for 24 aspect and elevation segments. By the way, stability information, mostly not modelled, has been used in several previous studies, even back in some early work on numerical forecasting in the 1990ies – and was shown to be important.

We thank Jurg Schweizer for both edition and useful comments that help us improve the paper. We answer point by point below. The original review is reported in green and associated with the answers in black. Quotations from the paper are in italic font and proposed changes in purple italic font.

We fully agree that the selected dataset can contain bias and is a specific representation of the avalanche activity. Before publishing the paper, we ensure that the main results are not largely affected by the main drawbacks we identify in the discussion phase (threshold effect, Haute-Maurienne specific climate, etc.). We also discuss the advantages and limits of the dataset in section 4.3. In particular, we underline that the trained model is specific to Haute-Maurienne due to the dataset used (line 424 and following), even though we believe the method can easily be transferred to other areas.

We adapted the title not to word it as a question.

**Specific comments**

> Below you will find some more specific comments:
>
> - Lines 13-14: I think it would be valuable to show how the model would perform with a simple target variable (not considering 24 aspect and elevation segments). Moreover, I think "cutting-edge" should not be related to data here.

We removed *cutting-edge*.

- Lines 20-21: I recommend rewording. Long-term and short-term are somewhat oddly used here. I suppose you refer to hazard assessment in the context of hazard mapping vs. avalanche forecasting. Hazard mapping and avalanche forecasting are both mitigation measures having a long-term and a short-term effect, respectively.

We now use *mapping* and *forecasting*.

- Line 24: I am not sure whether you focus on forecasting or prediction.

As forecasting has a connotation of prediction on future (defined in the first paragraph), and we work on data of the past, we then use the term prediction in the rest of the paper, except when dealing with practical use. Moreover, the term prediction is a common term for the machine learning community.

- Line 34: Please cite the corresponding peer-reviewed publication rather than a magazine article (Schweizer and Jamieson, 2007).

Replaced.

- Line 53: Some early models that used stability information were described by Schweizer et al.

  (1994) and Schweizer and Föhn (1996).
- Line 58: Schirmer et al. (2009) used output variables from a numerical snow cover model as input for statistical forecasting.
- Line 65: It has been shown that the type of data (input variables) can at least be as important as the method applied. For instance, regardless of the method, with meteorological data as input only, the performance is always limited (e.g., Schirmer et al., 2009)

Thanks for pointing out these references. We added two sentences in the introduction to acknowledge these contributions:

*The first machine learning models [Navarre et al., 1987; Buser, 1989] mainly rely on meteorological observations, simple snow observations and avalanche records. The use of modelled snow information was therefore developed to complement or replace observations [e.g. Schirmer et al., 2009; Sielenou et al., 2021] and expert analyses were introduced to provide appropriate variables [Schweizer and Fohn, 1996].*

- Lines 66-67: Suggest rewording to better reflect the aim.

We reworded the sentence by adding the idea that we compare the impact of using stability indices as predictors on the model performance.

- Lines 67-68: Suggest rewording so that the three different types of input data become more obvious.

We reworded the description of the three types of data.

- Line 79: In your replies to the reviewers' comments, you point out many uncertainties related to the dataset. Hence, I wonder whether you can state that the avalanche observations are particularly reliable.

We removed the sentence.

- Lines 97-98: As you describe above, EPA includes primarily large natural avalanches in selected path. It is questionable whether these data describe the overall avalanche activity in the area. There is much more potential avalanche terrain in the area and almost always when there are large or very large avalanches, there will be many small and medium- sized avalanches as well. On the other hand, I suspect, there are also numerous days when no very large natural avalanches occurred, but still many small to medium-sized natural avalanches at higher elevations. Hence, what is observed and recorded in EPA cannot provide a complete picture of natural avalanche activity. All these additional smaller natural avalanches are very likely also relevant for operational avalanche forecasting. Hence, in conclusion, I suggest you reword and specify the statement.

We removed the end of the sentence to keep only the idea of the reduction of the importance of the observation threshold in the specific case of Haute-Maurienne: *Besides, the steep topography of Haute-Maurienne reduces the*

*effect of the observation threshold as most avalanches flow far downslope, close to the valley floor.*

- Line 133: I recommend you add "for a given depth" or "a given layer interface"

We added *for a given layer interface*, as suggested.

- Line 146&147: I recommend replacing humid by wet or moist.

We prefer the generic term *humid* as it does not directly refer to the liquid water content scale from Fierz et al. (2009), especially as we vary the threshold to define a layer as humid. Wet or moist would each one be inappropriate with some thresholds.

- Line 156: As far as I remember, Reuter et al. (2022) have recently applied a time-dependent index to describe natural failure initiation.

As far as we know, it is a derivation of Conway and Wilbour (1999) index. We added the reference as an illustration of further use of the work of Conway and Wilbour.

- Line 160: It is confusing to the reader that changes in snow depth are associated with stability indices. Hence, I suggest regrouping the input variables.

The change in snow depth is now grouped in the derivative group, which is obviously more consistent. We adapted the results accordingly. This change highlights the importance of the time-derivatives, which we now discuss.

- Line 166: I agree that the model resolution you select is more demanding. However, the question remains unanswered whether this additional sophistication represents an added value given the overall rather poor performance of the model from an operational point of view.

In this paper, we decided to work with the presented geometry (AE segments). The goal of the paper is not the evaluate this choice. However, we studied this point (not shown) and we were able to prove that this specific geometry provides an added value compared to a similar algorithm applied at the massif scale.

- Lines 174-175: I suggest using more descriptive variable category names, for instance, weather, snow and stability. Certainly, "Bulk" does not seem appropriate to me. Also, in Table 1, you refer to derivates, which seem to include snow depth changes (though you only refer to dry snow indices). Please clarify.

We reworded the "Bulk" group as "Simple snow".

- Lines 219-221: I suggest you also mention that the recall is, commonly in the forecasting context, called the probability of detection (POD) or hit rate. This would ease comparison with previous work.

We added this term.

- Lines 221-222: I suggest you also mention that the false positive rate is called the false alarm rate (FAR).

We added this term.

- Line 236: Random classification is usually associated with the diagonal in the ROC diagram.

Edited.

- Line 254: elevation

Corrected.

- Lines 257-258: The example you provide is probably one of the most prominent avalanche winters in your dataset. There were three, well-known major avalanche cycles in late January and February in 1999. It would certainly be interesting to know the performance in such a "catastrophic" winter. In addition, are these model results obtained with the model trained with all input variables?

We now precise that all the variables were used for this figure.

- Lines 262-263: Please adapt the caption in Figure 4a to the description provided here.

Adapted

- Line 270: It may be worth mentioning that sensitivity (75.3%) and specificity (100-23.6%) are similarly good, which is nice, but that due to the unbalanced dataset the precision is low (3.3%).

We introduced the specificity and added the comment you suggested.

- Lines 278-279: The text here, in the revised manuscript, is unchanged, but Figure 5 (previously Figure 4) changed, i.e. the importance scores are different now. Are all changes in Figure 5 reflected in the text?

The different groups were slightly adapted but the results did not change. We carefully checked the whole paragraph.

- Line 280: In Figure 5 it is indicated that there are 25 variables from the dry snow stability group whereas in the text you refer to 30.

This was corrected.

- Line 294: Here you denote snow depth and its derivatives as "bulk", in contrast to the variable categories described in Table 1. Please clarify and/or improve Table 1.

Corrected. It concerns snow depth and new snow depth.

- Lines 313-314: What means "e.g. 2021"? By the way, both studies are published now. Please update the references (Mayer et al., 2022; Pérez-Guillén et al., 2022).

Updated and corrected.

- Line 336. I am not aware of any country who explicitly specifies the avalanche danger in 24 aspect and elevation segments.

We reworded the sentence to avoid this confusion. "This spatial resolution enables to capture the spatial distribution of the expected avalanche activity in one region. This latter information is crucial to evaluate and describe the avalanche danger at regional scale (Morin et al., 2019)."

- Lines 347ff: It seems odd to me not to consider snow depth and in particular new snow depth as meteorological variables. In any case, you cannot conclude that meteorology is irrelevant and that this contrasts with other studies that probably all considered new snow depth. Moreover, it is doubtful that the alleged difference stems from modelled vs. measured new snow depth. Most readers would probably agree that new snow depth (precipitation) is a meteorological driver of avalanche danger. Hence, it seems that your conclusion depends on your choice of categories and cannot be generalized.

This sentence supports the statement *Meteorological information only was insufficient.* Meteorological variables include snowfall, rainfall, temperature and wind speed and direction only (Table 1) and do not contain any snow depth for instance. Moreover, we do not say that meteorology is irrelevant in the general case, we state that it is insufficient to predict avalanche activity at high spatio-temporal resolution. We fully agree that it may be sufficient at larger scales and most studies also include other information, such as bulk snowpack information. We now insist in the first pointed sentence: *Meteorological information only was insufficient to predict avalanche activity with our method.*

- Line 353: Isn't new snow depth the significant variable rather than snow depth?

We added *snow depth and new snow depth*.

- Lines 559-360: I am not sure how this last sentence refers to the importance of new snow depth.

We are not sure to have understood the comment. For sure, evaluating new snow depth (e.g., in kg/m2) does not require a snow model but we show that adding variables (e.g. snow depth variations) derived from the full stratigraphy simulated by a snow model improves the prediction score.

- Lines 371-373: As far as I remember van Herwijnen et al. (2016) and van Herwijnen et al. (2018) showed that dry-snow avalanche activity was correlated with snowfall on times scales of 2 or more days, while for wet-snow avalanches the correlation was shorter and with energy input. Hence this seems to be in contrast with what you describe. However, I agree with your statement in the following sentence.

It is an interesting comment and more work is required to investigate this point. However, a comparison of the results of the two studies are not straightforward as we present results on time-derivatives of stability indices whereas van Herwijnen et al. uses correlations on snowfall and energy-related variables. The snowpack model in-between also accumulates information on mass and energy balance on longer periods and this finally influences the stability indices. Moreover, the results given by Figure 5 have to be handled with care, as variables are highly correlated. For these reasons, we prefer to only generally check that the overall time scales are coherent with the underlying processes and do not go further in the interpretation. In the future, we will work on the reduction of the number of input variables. This will save computational time but also allow more interpretation of the variables importances, especially if we are able to select a set of variables that are not too much correlated. We keep in mind this comment to compare with the results of van Herwijnen et al. in further research steps.

- Line 383: Previously you referred to 2779 events, not 2518. Please clarify.

We corrected the sentence for the 2779 events as it refers to the number of observed avalanches, whereas 2518 corresponds to the number of avalanche situations once combined by aspects and elevation sectors (avalanche situations).

- Lines 384-285: Please refer to my previous comment on "the representative screenshot of the overall avalanche activity". In addition, I am not sure I understand why in Haute Maurienne the scarcity of reported avalanche events is not a problem.

The scarcity of observation is a challenge. We used different techniques to balance the dataset for the machine learning method. We already have nearly two orders of magnitude between the number of avalanche and non-avalanche situations. But in some lower massifs with large forested areas, the ratio could be dramatically lower. In this case, we do not expect the balancing mechanism we use to be sufficient to provide useful results. We add *as our balancing methods may become insufficient* to explain the difference with Haute-Maurienne.

- Line 412: I am not sure I understand what you mean here with "the interest of physics".

We changed to *The impact of using physically-based indices of snow stability as predictors of avalanche activity instead of simpler variables*.

- Lines 419-420: Please reword.

We split the sentence. Hope this is clearer: *These alternative statistical methods could be further compared to our random forest approach. It may provide improvements in the prediction scores or strengthen our results on the effectiveness of combining snow physics and machine learning for predicting avalanche activity.*

- Line 429: In my understanding "avalanche prediction" means to predict the exact time and location of a single avalanche event. I think so far, we rather forecast avalanche activity.

The wording was confusing, we added the precision *avalanche activity prediction*.

- Lines 434-435: I suggest rewording the statement: the combination proves to be valid.

We reworded as *proves to be useful for avalanche activity prediction*.

- Line 435: I guess new snow depth was a significant variable, not snow depth.

Corrected.

- Line 436: As far as I remember, Jamieson and co-workers have shown that stability indices, though derived from measurements (not modeled), were valuable inputs for forecasting (e.g., Zeidler and Jamieson, 2004).

We added the reference.

- Line 439-440: While I agree that snow cover models and thereof derived stability information is valuable, I recall that your model is not that great in identifying avalanche prone situations. The precision is low. Please consider rewording.

We reworded the sentence, which now reads: *Our results also underline the interest of physically-based snow cover models and stability indices for identifying avalanche-prone conditions.*

- Line 443: I suggest deleting "cutting-edge". The random forests method has become a rather standard tool over the course of the last decade. In addition, I am not sure you can call the data cutting edge.
- Line 446: Similarly, not convinced the stability indices are that cutting-edge.

We removed both *cutting-edge*.

**References**

Mayer, S., van Herwijnen, A., Techel, F. and Schweizer, J., 2022. A random forest model to assess snow instability from simulated snow stratigraphy. The Cryosphere 16(11): 4593-4615.

Pérez-Guillén, C., Techel, F., Hendrick, M., Volpi, M., van Herwijnen, A., Olevski, T., Obozinski, G., Pérez-Cruz, F. and Schweizer, J., 2022. Data-driven automated predictions of the avalanche danger level for dry-snow conditions in Switzerland. Nat. Hazards Earth Syst. Sci., 22(6): 2031-2056.

Reuter, B., Viallon-Galinier, L., Horton, S., van Herwijnen, A., Mayer, S., Hagenmuller, P. and Morin, S., 2022. Characterizing snow instability with avalanche problem types derived from snow cover simulations. Cold Reg. Sci. Technol., 194: 103462.

Schirmer, M., Lehning, M. and Schweizer, J., 2009. Statistical forecasting of regional avalanche danger using simulated snow cover data. J. Glaciol., 55(193): 761-768. Schweizer, J. and Föhn, P.M.B., 1996. Avalanche forecasting - an expert system approach. J. Glaciol., 42(141): 318-332.

Schweizer, J. and Jamieson, J.B., 2007. A threshold sum approach to stability evaluation of manual snow profiles. Cold Reg. Sci. Technol., 47(1-2): 50-59.

Schweizer, M., Föhn, P.M.B., Schweizer, J. and Ultsch, A., 1994. A hybrid expert system for avalanche forecasting. In: W. Schertler, B. Schmid, A.M. Tjoa and H. Werthner (Editors), Information and Communications Technologies in Tourism, Innsbruck, Austria, 12-14 January 1994. Springer Verlag Wien, New York, pp. 148-153.

van Herwijnen, A., Heck, M., Richter, B., Sovilla, B. and Techel, F., 2018. When do avalanches release: investigating time scales in avalanche formation. In: J.-T. Fischer et al. (Editors), Proceedings ISSW 2018. International Snow Science Workshop, Innsbruck, Austria, 7-12 October 2018, pp. 1030-1034.

van Herwijnen, A., Heck, M. and Schweizer, J., 2016. Forecasting snow avalanches by using avalanche activity data obtained through seismic monitoring. Cold Reg. Sci. Technol., 132: 68-80.

Zeidler, A. and Jamieson, J.B., 2004. A nearest-neighbour model for forecasting skier-triggered dry- slab avalanches on persistent weak layers in the Columbia Mountains, Canada. Ann. Glaciol., 38: 166-172.

**Answer to Simon Horton review**

Léo Viallon-Galinier        Pascal Hagenmuller        Nicolas Eckert

**General comments**

> I appreciate the thoughtful revisions to this manuscript which have clarified many of my initial concerns. Many of the methodological choices are now clearer and better put the results into context. In particular, the explanation of how of a correctly predicting avalanche activity in specific elevation and aspect terrain classes is more difficult that predicting activity at a regional scale, is much clearer. With these rationales clearer I still have some concerns about how the variable groupings impact the results and a few suggestions to strengthen some arguments. Thanks, it was an enjoyable and interesting manuscript to read.

We thank the reviewer for his positive feedback and hope that the following adjustments match the reviewer suggestions and improve the paper. We answer point by point below. The original review is reported in green and associated with the answers in black. Quotations from the paper are in italic font and proposed changes in purple italic font.

**Specific comments**

- **Variable groupings**. I find the limited choice of meteorological variables misleading, especially when this group performs similar to a random classifier. First, not all the meteorological inputs appear to be included, especially ones that would specifically improve the prediction at the spatial resolution investigated. For example, solar radiation and wind direction should be primary drivers of snowpack variability across different aspects and precipitation, temperature and wind speed should be driving variability across elevations. The resulting snowpack properties on different terrain classes are ultimately a product of the meteorological inputs, and by not including these in the analysis it is not a representative comparison of a model with and without stability indices. With these groupings some of the predictive skill being attributed to stability indices does not truly reflect the added value of a snowpack model. Similarly, including snow depth change in the stability group may also inflate the importance of stability indices. First, based on many previous studies this is expected to be one of the most important predictors of avalanche activity. Second, it would make more sense to consider this a meteorological variable or, specifically in this study, a derivative of a bulk variable. While I understand the argument that the current structure shows how stability indices aggregate the information in a way that explains the dataset well, the goal of the study suggested by the title is to ask "does it help?", which I think requires a precisely thought out structure that compares the ability to predict avalanche-situations with and without stability information.

We changed the title according to J. Schweizer proposition.

On the input variable, we selected a set of variable that is not uncommon but necessarily arbitrary. In particular, solar radiations are not included as even though it is quite important for the snowpack, it is not commonly measured nor used directly as a meteorological variable. However, wind speed and direction are included.

We fully agree that the snowpack is the result of the evolution under the meteorological conditions, but we here consider that this synthesis is the goal of the snow cover model. We then do not re-do the job of the snow cover model by introducing additional variables. The same way, we assume that stability indices are a good summary of the complex stratigraphy represented by the snow cover model. It is also possible to consider that all information is produced by the snow cover model and use all output variable as predictors. The same way, it is possible to use only the input variables of the snow cover model (meteorological variables) as once again, all the final information comes from this early data. The goal of this study is to compare three common levels of information (meteorological data, bulk snowpack information, stability indices). This separation is somehow arbitrary but not fully original as it

roughly corresponds to the main classes of McClung and Schaerer, 1993.

- **Variable importance**. The way the results of the Gini importance are presented in Fig. 5 make it difficult to follow the discussion in lines 277-286, because individual variables are discussed in the text but only the group values are shown in the figure. For example, although snow depth is reported to be the most important, its value is not reported in the text and the group of dry snow stability indices collectively have more importance in the figure. I think these results could be presented in a clearer and more consistent way. In some ways this relates to the previous comment about some arbitrary groupings. Perhaps presenting the Gini importance values for every variable in an appendix would help, perhaps sorted by importance within each group.

We do not believe that Gini importance for individual variable have any sense. As explained in section 2.6.4 and reminded in the presentation of the results, the analysis of such variable importance is only mathematically justified when variables are independent. This is absolutely not the case here. Therefore, we keep Figure 5 only because this method is very commonly used with random forests [e.g. Sielenou et al., 2021; Mayer et al., 2022] and allow for a first easy overview of the variable importance but the values have to be handled with care as there is a lot of correlation between variables and even between the groups of variables. For instance, two variables perfectly correlated will have roughly the same Gini importance with a value corresponding to half of the Gini importance of one of the two variables if the other variable was removed from the dataset.

To improve readability, we added the part of the sentence that mentioned the detail on variables inside a group.

- **Oversample low snow depth days**. Initially from a forecasting perspective the 10 cm threshold to remove non-avalanche situations seems overly conservative. My concern is the importance of snow height would dominate over any influence of stability indices, especially in a dataset dominated by full path avalanches. After reviewing the results in more detail, I am now satisfied with this concern by seeing that the model with stability indices alone, without snow depth, perform at a similar level and thus must capture the important influence of snow depth some way. I'm wondering if this specific point should be emphasized more to highlight that the impact of threshold snow depth can be captured within the set of stability variables.

We believe that we have an even more large conclusion when we state that *The introduction of stability indices and time-derivatives could help identify avalanche-prone situations with machine learning models. This group of variables also gathers a great deal of information as it nearly replaces the information from other variables.* (Section 4.2).

- Impact of aspect-elevation resolution on performance. Can you explicitly state why the aspect- elevation resolution leads to lower performance due to more precise resolution? I assume it's because it is harder to predict the correct aspect and elevation of an avalanche than predict an avalanche anywhere in a region. Providing a direct explanation of this in the discussion would strengthen the argument that the some of the low performance metrics are justified.

Exactly, the difficulty comes from our evaluation at the aspect-elevation scale, which means that for a given avalanche situation, we have to predict both the avalanche or non-avalanche situation but also in the correct aspect and elevation band. We edited the corresponding paragraph to explicitly explain why aspect-elevation resolution is more demanding with an example:

*Our model predicts the probability that at least one avalanche occurs on a given day within a spatial unit corresponding to one elevation band (centred at 1800, 2400 and 3000m) and one aspect (among 8 aspects). This spatial resolution enables to capture the spatial distribution of the expected avalanche activity in one region. This latter information is crucial to evaluate and describe the avalanche danger at regional scale [Morin et al., 2019]. This prediction goal is more demanding than a prediction at larger scales, as generally used in previous studies. Indeed, prediction at aspect-elevation resolution implies to correctly predict the avalanche activity for each aspect and elevation band and not globally at a larger scale. For instance, if one avalanche occurs one day, it implies to identify that we have one avalanche situation but also in which aspect and elevation sector to be considered a success. An avalanche predicted in an other elevation or aspect will be considered as one false negative (in the elevation-aspect it really occurred) and one false positive (in the elevation-aspect it was predicted). It inevitably leads to lower performances for similar models but provides more precise information about the spatial distribution of the avalanche hazard [Statham et al, 2018].*

**Technical comments**

- Line 97-98: Can you clarify what is meant by "observation threshold"? Perhaps this could be clarified by being more specific in the line 94 with "run-out reached a certain threshold distance" and then in line 97-98 sentence make it clear "although the observation data only includes avalanches that reach the threshold distance, the dataset provides an overall representative indication of avalanche activity in the area because the steep topography of the Haute-Maurienne causes most avalanches to reach this distance…"

We now explain that it is an *run out threshold (defined for each avalanche path)*.

- Table 1: Why is snow depth change in stability?

We clarified the grouping by associating the change in snow depth in the derivative group and adapted the results accordingly.

- Line 196: Is there a word missing in "goal is not to substitute to the machine learning algorithm"? The sentence is unclear.

We reworded the sentence to make it clearer: *Note that we chose this conservative threshold to remove very obvious non-avalanche situations from the dataset (no snow in the starting zone means no avalanche). We do not expect this threshold to be optimal as this is the goal of the training phase of the machine learning algorithm.*

- Line 278-279: Why are the Gini importance values not reported for these most importance variables, but are reported for subsequent groups? The reporting of these results is inconsistent and difficult to interpret from Fig. 5.

Please refer to the main comment. We use groups that are a little bit more independent than individual variables. We do not believe that Gini importance for individual variable have any sense. As explained in section 2.6.4 and reminded in the presentation of the results, the analysis of such variable importance is only mathematically justified when variables are independent. This is absolutely not the case here. Therefore, we keep Figure 5 only because this method is very commonly used with random forests [e.g. Sielenou et al., 2021; Mayer et al., 2022] and allow for a first easy overview of the variable importance but the values have to be handled with care as there is a lot of correlation between variables and even between the groups of variables.

- Sect. 4.2: It would be interesting to report which specific stability indices had the highest relative importance. Why are these not presented? This could be interesting for future reach into stability indices.

We fully agree with this comment. However, we do not think that a table like Figure 5 could answer the question as some stability indices are highly correlated. Moreover, we think that this goes beyond the main goal of this paper. However, we plan to pursue this work by determining a minimal set of input variables. In this process, we will have to identify the most relevant stability indices. The second method will be necessary. However, it is not realistic to test all the variable combinations with this method. We thus plan to combine our knowledge of the different processes represented by stability indices as well as previous studies to provide a first selection before doing evaluations like in Figure 6.

---

## Author Response (AR3)

**Answer to Editor**

Léo Viallon-Galinier          Pascal Hagenmuller          Nicolas Eckert

> Many thanks for your revisions.
>
> As you an see the reviewers acknowledge that you have adequately addressed their comments, although they remain somewhat unconvinced on certain aspects such the target resolution with 24 AE segments and the relevance of new snow depth. I do not require that you make fundamental changes, different opinions are Ok in my view. Nevertheless, I will give you the opportunity to further improve the manuscript by possibly considering their comments. I also made suggestions, see the annotated manuscript. After you resubmit I will accept the paper.
>
> Best regards, Jürg Schweizer.

We are really thankful to Jürg Schweizer for the time dedicated to this paper for the edition process and for providing constructive feedback and useful suggestions.

We took into account the different comments reported in the attached PDF, as well as the reviewer's comments to revise the manuscript. We agree we made some assumptions in this study that will need to be exceeded in the future, with the help of different methods or alternative datasets for instance. However, we hope we correctly identify them, with the help of the reviewers, and acknowledge clearly these limitations in the text.

**Answer to Simon Horton comments**

Léo Viallon-Galinier        Pascal Hagenmuller        Nicolas Eckert

**General comments**

> The revised manuscript has resolved many of my concerns and clarified the objective and limitations of the study.

We are really thankful to Simon Horton for the time dedicated to review this paper. The constructive comments helps improving the paper during the different review steps.

> I still find the presentation of variable importance with the Gini values in Fig. 5 misleading. The bars show the total importance value for a group of variables, but then the interpretation in the text accounts for the number of members in each group, which results in variables discussed as being "important" (e.g., snow depth variations, weak layer depths) having smaller bars in the figure than groups of variables that are not discussed. Importance values are reported in the text for some but not all groups, such as 13.6% for dry snow stability and 3.6% for weak layer depth, but no values are reported for the new snow amounts which are discussed as being important throughout the manuscript. It would be nice for the manuscript to more clearly present these results, such as modifying Fig. 5 to show an average value for a group (importance value for a group divided by the number of variables), reporting more numbers in the text to make stronger comparisons, or perhaps other methods used to make the claim new snow depth was important.

We decided to provide the Gini importances because this is a very common approach with the Random Forest (RF) method. However, in the present case, the interpretation of the importance of variables provided by this method have to be handled with care because the computation of Gini importance from RF method is only valid for independent predictors, which is absolutely not the case here. We remind this in different paragraphs of the manuscript (section 2.6.4 and 3.3) as well as in the different review answers. Then, we only provide these Gini importance for illustration of the results at higher resolution than permitted by the other method, to check the variable importance regarding to current knowledge and to highlight the limitations of this method for the estimation of variable importance.

On one hand, the summed importance variable for each group, could be seen as the overall importance of the group, if we consider the different groups are sufficiently independent. On the other hand, the number of variables gives an idea of the complexity of the group (increase in complexity in the RF trees, as well as the complexity of the interpretation due to the number of highly correlated variables). Therefore, we believe that both are of interest. In any case, neither the absolute values of Gini importance nor the average value is fully mathematically justified as soon as variables are not independent two by two.

Therefore, we keep Figure 5 only because this method is very commonly used with random forests [e.g. Sielenou et al., 2021; Mayer et al., 2022] and allow for a first easy overview of the variable importance but the values have to be handled with care as there is a lot of correlation between variables and even between the groups of variables. We are now more precise on this goal and the objective of the part of the result section concerning the Gini importance :
*However, absolute values have to be taken with care as this analysis method is strictly valid only when the different variables are independent, which is far from the case we have here. We thus provide this analysis to check the main results according to previous knowledge and because of the popularity of this method, but the detailed results are of limited interest due to the presented limitations of this method in our study case.*

**Technical comments**

> I have two other technical comments.
>
> - Line 157: I assume this addition is for wet avalanches only? ". . . in a potential wet avalanche."

As these two variables are relevant for both dry and wet snow avalanche activity, we moved this sentence to the introductory paragraph.

> - Why are not all the scores described in Sect. 2.3.2 listed in Table 3? Would be helpful to move all equations from the text into the table (specificity, balanced precision, etc.).

We moved all equations into Table 3 as suggested.

**Answer to Frank Techel comments**

Léo Viallon-Galinier        Pascal Hagenmuller        Nicolas Eckert

> Dear authors

> Thank you for revising the manuscript.

We are really thankful to Frank Techel for the time dedicated to review this paper and for providing a very interesting, constructive and helpful feedback.

> Following are some minor typos or recommendations:
>
> - l48: change "become then" to "became"
> - l117: there is an "i" too much
> - l159: I am not sure whether two plurals are correct in "stability indices values".
> - l176: I suggest adding a reference to the description of the filtering in Sect 2.2. at the end of the brackets ", as explained in Sect. 2.2", or similar
> - l177: replace "ones" with "situations"
> - l415: add an "s" to "path"

We took into account all these typos/recommendations in the revised manuscript.

> - l59/60: consider removing "could increase the interpretability of the algorithm results and"

For the sentence line 59 and 60 we believe that such preprocess of data to produce relevant stability indices could both reduce the complexity of the algorithm and therefore increase the interpretability.

> Kind regards,

> Frank Techel